# Neutrophil extracellular traps released by neutrophils impair revascularization and vascular remodeling after stroke

Lijing Kang [1,2], Huilin Yu [1,2], Xing Yang[1], Yuanbo Zhu[1], Xiaofei Bai[1], Ranran Wang[1], Yongliang Cao[1], Haochen Xu[1], Haiyu Luo[1], Lu Lu [1], Mei-Juan Shi[1], Yujing Tian[1], Wenying Fan [1✉] & Bing-Qiao Zhao [1✉]

Neovascularization and vascular remodeling are functionally important for brain repair after stroke. We show that neutrophils accumulate in the peri-infarct cortex during all stages of ischemic stroke. Neutrophils producing intravascular and intraparenchymal neutrophil extracellular traps (NETs) peak at 3–5 days. Neutrophil depletion reduces blood-brain barrier (BBB) breakdown and enhances neovascularization at 14 days. Peptidylarginine deiminase 4 (PAD4), an enzyme essential for NET formation, is upregulated in peri-ischemic brains. Overexpression of PAD4 induces an increase in NET formation that is accompanied by reduced neovascularization and increased BBB damage. Disruption of NETs by DNase 1 and inhibition of NET formation by genetic ablation or pharmacologic inhibition of PAD increases neovascularization and vascular repair and improves functional recovery. Furthermore, PAD inhibition reduces stroke-induced STING-mediated production of IFN-β, and STING knock-down and IFN receptor-neutralizing antibody treatment reduces BBB breakdown and increases vascular plasticity. Collectively, our results indicate that NET release impairs vascular remodeling during stroke recovery.

[1] Department of Translational Neuroscience, Jing'an District Centre Hospital of Shanghai, State Key Laboratory of Medical Neurobiology and MOE Frontiers Center for Brain Science, Institutes of Brain Science, Fudan University, Shanghai 200032, China. [2]These authors contributed equally: Lijing Kang, Huilin Yu. ✉email: wenyingf@fudan.edu.cn; bingqiaoz@fudan.edu.cn

Stroke is one of the most common causes of long-term disability with limited therapeutic options[1]. Recent evidence suggests that neovascularization after brain injury is functionally important for endogenous repair process[2,3] and blockade of angiogenic response worsens outcomes after cerebral ischemia[4]. Therefore, promoting faster revascularization is of great therapeutic interest for treating a wide range of central nervous system diseases. To achieve this goal, an improved molecular understanding is required of the mechanisms underlying vascular regrowth in the injured brain.

Neutrophils were traditionally considered as the first line of innate immune defense against microbes[5,6]. In addition to playing a role in bacterial killing, activation of neutrophil causes the release of nuclear and granular contents to form extensive web-like structures of DNA (neutrophil extracellular traps, NETs)[7,8]. NETs contain double-stranded DNA, histone, and granule proteins including neutrophil elastase, cathepsin G, and myeloperoxidase (MPO)[9]. These NETs have been associated with autoimmune disorders[10], cardiovascular and pulmonary diseases[11,12], inflammation[13], and thrombosis[14,15]. Blood-derived neutrophils and release of NETs have been identified in the brains of patients with ischemic stroke and in corresponding animal models[16,17]. However, whether NETs contribute to favorable or poor outcomes during stroke recovery remains unclear. Furthermore, the role of neutrophils in cerebral ischemia has also been challenged. A recent study shows that both polymorphonuclear granulocytes and NETs were absent in the brain parenchyma after ischemic stroke in animals and humans[18].

The peri-infarct cortical areas have been previously shown to be critical for functional recovery in animal stroke model[19,20]. In stroke patients, increased vascularization in these areas is also correlated with longer survival[21]. In this study, we detect elevated levels of circulating DNA and show neutrophil-dependent NET formation inside the blood vessels and cerebral parenchyma in the peri-infarct cortical areas that peaked at 3–5 day after cerebral ischemia. We also demonstrate that target neutrophils and NETs improve cerebrovascular remodeling and functional recovery during the delayed phases after stroke.

## Results

**Neutrophils damage vascular remodeling after stroke.** We subjected mice to cerebral ischemia and analyzed the brains at 1, 3, 5, 7, and 14 days. Using an anti-lymphocyte antigen 6 complex locus G (Ly6G) antibody, we found elevated levels of neutrophil that peaked at 3 days and persisted to at least 14 days in the ischemic cortex after stroke (Fig. 1b, c and Supplementary Fig. 1a), as reported[22]. We then found a marked 5.3-fold increase in total content of the neutrophil enzyme MPO in the cortical areas at 3 days after stroke (Fig. 1d). Immunostaining showed that neutrophils were visualized throughout the peri-infarct cortex at 3 days but not after sham surgery (Fig. 1e). Neutrophils were observed inside blood vessels, adhering to vessels or migrating inside the parenchyma (Fig. 1f). The extravasation of neutrophils in the brains of ischemic mice was further confirmed using in-vivo multiphoton microscopy (Fig. 1g). Ly6G-labeled neutrophils abundantly infiltrated into the peri-infarct cortex at 3 days, whereas these cells were almost undetectable in the sham group. We also saw a higher percentage of circulating neutrophils in peripheral blood at 3 days after stroke as determined by flow cytometry (Fig. 1h).

Leukocyte infiltration contributes to blood–brain barrier (BBB) damage after brain injury by producing reactive oxygen species, proteases, and proinflammatory mediators[23–25]. To investigate the role of neutrophils on cerebrovascular permeability during delayed phases after stroke, we depleted neutrophils in mice using an anti-Ly6G antibody for 14 days. Flow cytometry showed an 80% reduction in the number of blood neutrophils (Supplementary Fig. 2a, b) in mice treated with anti-Ly6G antibody. Peripheral blood counts also indicated that anti-Ly6G treatment reduced neutrophil counts in the blood by ~80% when compared with the isotype control group (Fig. 2a). White blood cell counts were also reduced (Fig. 2b), whereas red blood cell, platelet, monocyte, and lymphocyte counts were not significantly affected by anti-Ly6G treatment (Supplementary Fig. 2c-f), as reported[26,27]. Correspondingly, immunostaining analysis indicated that the number of infiltrating neutrophils in the ischemic brain was significantly less in anti-Ly6G-treated mice than in control IgG-treated mice (Fig. 2c). Multiphoton microscopy analysis of intravenously (i.v.) injected fluorescein isothiocyanate (FITC)-dextran revealed a significant increase in vascular leakage in the peri-infarct cortical areas at 14 days after stroke compared with sham-operated brains (Fig. 2d, e), as we previously reported[28]. Mice treated with anti-Ly6G antibody exhibited a 39.3% reduction in BBB permeability compared with mice treated with control IgG. Extravascular accumulation of circulating IgG was also reduced in the brains of mice treated with the anti-Ly6G antibody (Fig. 2f, g). We then investigated the effect of neutrophils on neovascularization; the total length of brain capillaries in the peri-infarct cortical areas was analyzed at 14 days. Neutrophil-depleted mice showed a 29.8% increase in microvascular length compared with the control IgG-treated mice (Fig. 2h, i). Multiphoton microscopy revealed that neutrophil depletion caused a significant increase in the length of perfused cortical microvessels (Fig. 2j, k). There was no significant difference in infarct volume between anti-Ly6G-treated and control IgG-treated mice at 14 days (Supplementary Fig. 2g, h). Together, these data indicate that neutrophils are important regulators of neovascularization and vessel function during stroke recovery.

**Stroke induces NET formation.** To test whether NETs are present in the circulation of ischemic mice, blood cells from mice at 3 days after stroke were stained for citrullinated histone H3 (H3Cit) and Ly6G. The results revealed a significantly higher number of neutrophils and H3Cit+ neutrophils in ischemic mice compared with sham-operated mice (Fig. 3a-c). In line with this, elevated levels of circulating DNA was found in the plasma from these mice (Fig. 3d). We then isolated neutrophils from the peripheral blood of these mice and incubated them with or without lipopolysaccharide (LPS) stimulation. We found that either unstimulated or LPS-stimulated neutrophils from ischemic mice showed a significant increase in H3Cit+ neutrophils and NET formation (Fig. 3e-g), indicating that neutrophils from mice subjected to stroke are primed to undergo NETosis.

As NETs can injure host tissue[29], we next asked whether NETs were produced in the ischemic brain and affect stroke outcomes. Western blot analysis of the ischemic cortex showed an increased amount of H3Cit that was most robust from 3–5 days (Fig. 3h, i), suggesting that NETs may play an important role during the delayed phases after stroke. Immunostaining revealed that the peri-infarct cortex was extensively labeled with H3Cit+ cells at 3 days (Fig. 3j). To identify which type of cells expressed H3cit after stroke, double immunofluorescence with confocal microscopy was performed on brain sections. This analysis revealed that H3Cit was colocalized with Ly6G-positive neutrophils, F4/80-positive macrophages/microglial cells, Iba1-positive microglial cells, NeuN-positive neurons, and glial fibrillary acidic protein (GFAP)-positive astrocytes (Fig. 3k). Importantly, 78.7% of the H3Cit-positive cells were Ly6G-positive neutrophils. H3Cit+ neutrophils were observed inside the blood vessels and cerebral parenchyma (Fig. 3l). Hematoxylin and eosin staining clearly indicated that DNA fibers were present in these areas

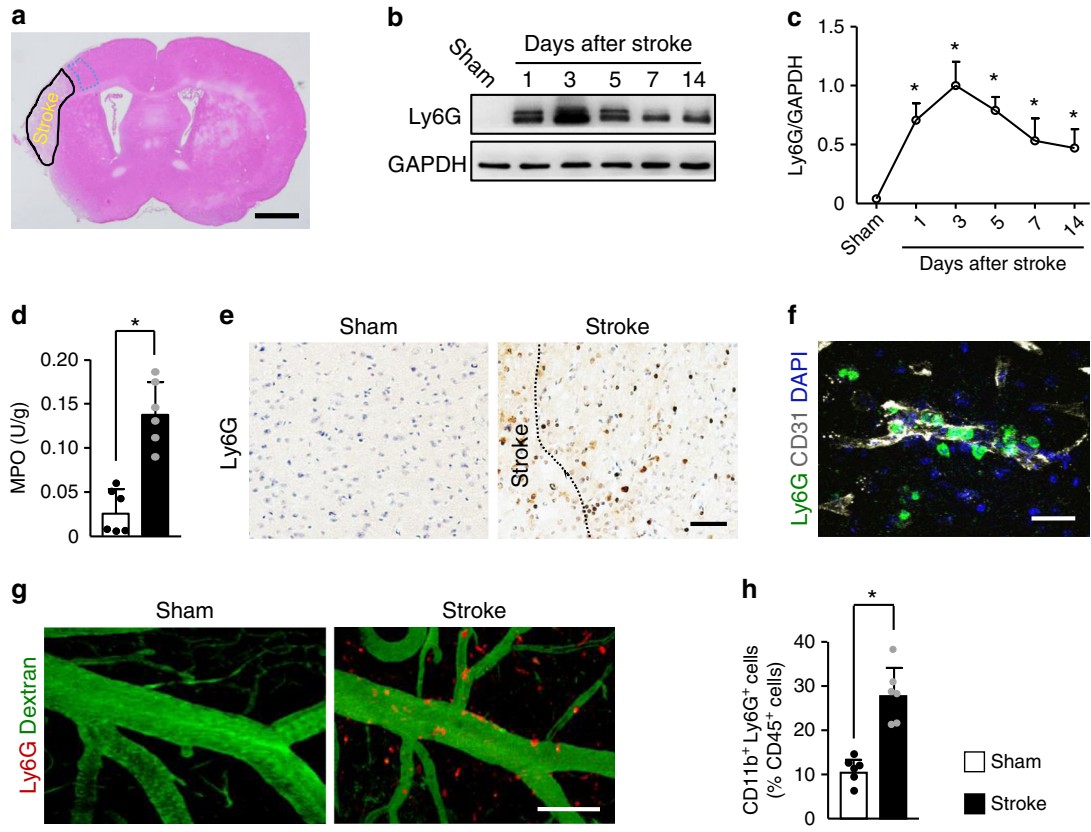

**Fig. 1 Neutrophils accumulate in the brain during all stages of ischemic stroke. a** Hematoxylin and eosin-stained coronal section shows the zone (broken blue line) used for measurements of IgG extravasation, capillary length, and in-vivo multiphoton microscopy in the peri-infarct cortical areas. Bar = 1 mm. Independent experiments are repeated at least three times. **b**, **c** Representative immunoblots of the time course of neutrophils appearance (**b**) and quantification of the amount of neutrophil (**c**) in the peri-infarct cortex of mice subjected to stroke, compared with sham-operated mice ($n = 5$). One-way ANOVA test was applied with *$P < 0.0001$ (Sham vs. 1d), *$P < 0.0001$ (Sham vs. 3d), *$P < 0.0001$ (Sham vs. 5d), *$P = 0.0003$ (Sham vs. 7d), *$P = 0.0015$ (Sham vs. 14d). **d** Quantification of MPO activity in the brain of mice at 3 days after stroke or sham operation ($n = 6$), unpaired two-tailed Student's $t$-test was applied with *$P = 0.0001$. **e** Representative images of Ly6G-positive neutrophils in the peri-infarct cortex of mice at 3 days after stroke, compared with sham-operated mice. Bar = 50 μm. Independent experiments are repeated at least three times. **f** Representative confocal images of Ly6G-labeled neutrophils (green) and CD31-positive microvessels (white) in the peri-infarct cortex of mice at 3 days. Nuclei were visualized with Hoechst. Neutrophils were observed within brain vessels and migrated into the parenchyma. Bar = 40 μm. Independent experiments are repeated at least three times. **g** Representative in-vivo multiphoton microscopy images of neutrophils (red) and cerebral angiopathy (green) in the peri-infarct cortex of mice at 3 days. Neutrophils were localized in brain vessels and the parenchyma. Blood vessels (green) were labeled by intravenous injection of FITC-dextran (MW = 2,000,000 Da). Neutrophils (red) were labeled by intravenous injection of PE-conjugated monoclonal Ly6G antibody. Bar = 100 μm. Independent experiments are repeated at least three times. **h** Flow cytometric quantification of neutrophils (CD11b+Ly6G+ cells) in peripheral blood at 3 days after stroke, expressed as percentage of total leukocytes (CD45+ cells) ($n = 6$), unpaired two-tailed Student's $t$-test was applied with *$P = 0.0001$. Data are presented as mean ± SD. Source data underlying graph **b**–**d** and **h** are provided as a Source Data file.

(Supplementary Fig. 3a). To confirm these observations obtained by histological analyses, we identified NET formation using in-vivo multiphoton microscopic imaging of i.v.-injected Sytox green. This analysis revealed that Sytox green-positive extracellular DNA fibers were not observed in the nonischemic cortex, whereas these fibers abundantly detected in the peri-infarct cortex at 3 days after stroke (Fig. 3m).

**Disruption of NETs enhances vascular remodeling after stroke.** We next investigated whether degradation of NETs with DNase 1 could improve vascular remodeling after stroke in mice. Treatment with DNase 1 did not affect the amount of neutrophils in the peri-infarct cortical areas (Supplementary Fig. 4a, b) but reduced H3Cit levels (Fig. 4a, b). Compared with the vehicle controls, administration of DNase 1 significantly reduced BBB permeability (Fig. 4c, d) and extravascular IgG deposits (Fig. 4e, f), and enhanced both Pdgfr-β+ and CD13+ pericyte coverage on

brain microvessels (Fig. 4g, h and Supplementary Fig. 4c, d). Vascular branches (Supplementary Fig. 4e, f), microvascular length (Fig. 4i), perfused capillary length (Fig. 4j, k), and tomato-lectin perfused vessels (Fig. 4l, m) were also increased in the brains of mice treated with DNase 1. Next, we investigated the importance of neutrophil NETs in vascular remodeling. Our results showed that treatment with DNase 1 in combination with anti-Ly6G antibody did not further improve neovascularization and BBB leakage compared with mice treated with anti-Ly6G antibody alone (Fig. 4n-p). These data indicate that DNase 1 primarily digests NETs generated by neutrophils, and that neutrophil NETs play a crucial role in vascular remodeling after stroke. However, in addition to neutrophil NETs, H3Cit in other cells may also contribute to the impaired vascular remodeling.

Next, we detected whether NETosis regulates vascular remodeling during repair processes. We found that injection of anti-Ly6G antibody beginning 7 days after stroke reduced extravascular IgG deposits at 14 days (Fig. 5a, b). Furthermore, we observed

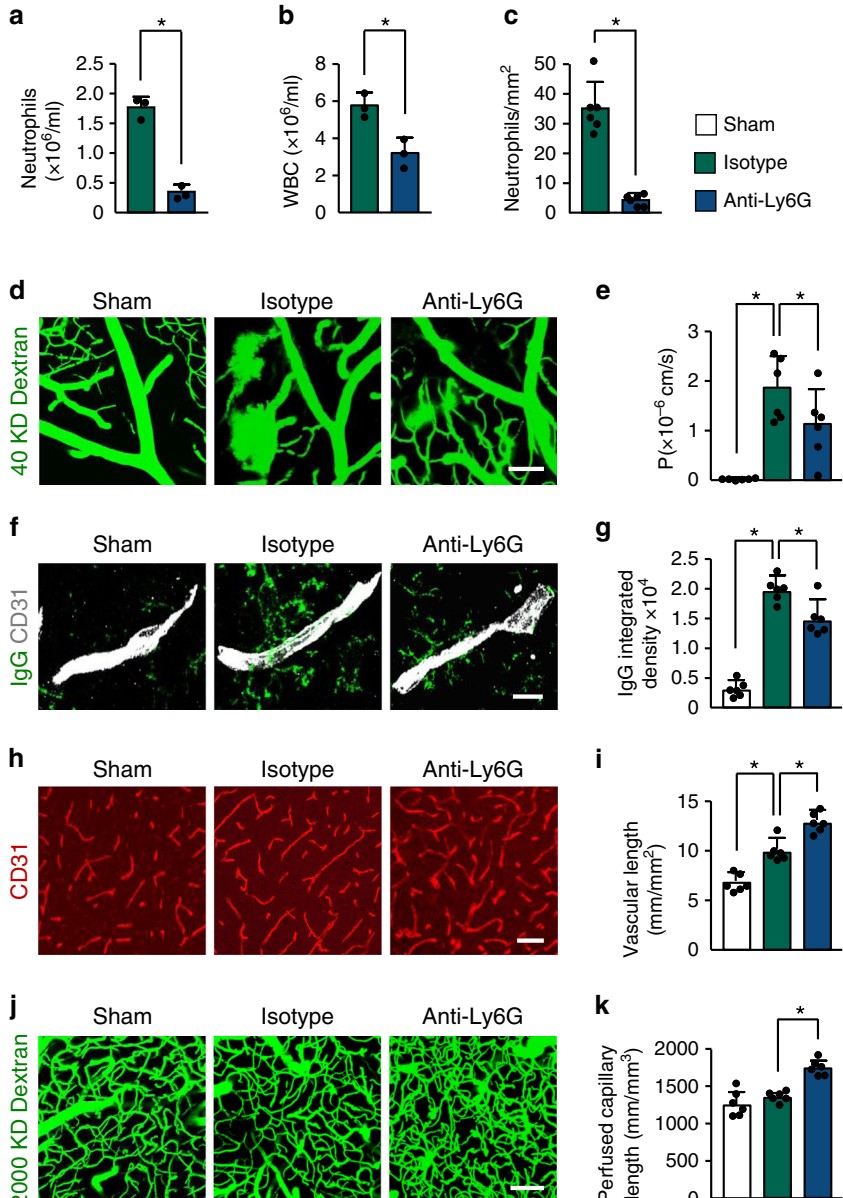

**Fig. 2 Neutrophil depletion reduces BBB breakdown and increases neovascularization after stroke. a**, **b** Neutrophil and white blood cell counts in peripheral blood at 14 days after stroke in mice treated with control antibody or anti-Ly6G antibody ($n = 3$), unpaired two-tailed Student's $t$-test was applied with $*P = 0.0003$ (**a**), $*P = 0.012$ (**b**). WBC, white blood cell. **c** Quantification of the number of neutrophils in the ischemic cortex at 14 days in mice treated with control antibody or anti-Ly6G antibody ($n = 6$ biologically independent experiments), Mann–Whitney test was applied with $*P = 0.0022$. **d**, **e** Representative in-vivo multiphoton microscopic images of intravenously injected FITC-dextran (MW = 40,000 Da; green) leakage in cortical vessels (**d**) at 14 days in sham-operated mice and ischemic mice treated with control antibody or anti-Ly6G antibody, and quantification of the permeability (P) product of FITC-dextran for each group (**e**) ($n = 6$). One-way ANOVA test was applied with $*P = 0.0001$ (Sham vs. Isotype), $*P = 0.0354$ (Isotype vs. Anti-Ly6G). Bar = 100 μm. **f**, **g** Representative confocal images (**f**) and quantitative analysis of IgG extravascular deposits (**g**) in the peri-infarct cortex at 14 days in sham-operated mice and mice treated with control antibody or anti-Ly6G antibody ($n = 6$). One-way ANOVA test was applied with $*P < 0.0001$ (Sham vs. Isotype), $*P = 0.0041$ (Isotype vs. Anti-Ly6G). Bar = 15 μm. **h**, **j** Representative confocal images (**h**) of CD31-positive microvessels and in-vivo multiphoton microscopy images of perfused cortical capillaries with intravenously injected FITC-dextran (MW = 2000,000 Da) (**j**) in the peri-infarct cortex at 14 days in mice treated with control antibody or anti-Ly6G antibody, compared with sham-operated mice. Bar = 50 μm (**e**) and 100 μm (**g**). **i**, **k** Quantification of microvascular density (**i**) and perfused capillary length (**k**) for each group ($n = 6$). One-way ANOVA test was applied with $*P = 0.0003$ (Sham vs. Isotype (**i**)) $*P = 0.0004$ (Isotype vs. Anti-Ly6G (**i**)), $*P = 0.0002$ (Isotype vs. Anti-Ly6G (**k**)). Data are presented as mean ± SD. Source data underlying graph **a**–**c**, **e**, **g**, **i**, and **k** are provided as a Source Data file.

significant increases in microvascular length (Fig. 5e, f) and perfused cortical vessels (Fig. 5i, j) in anti-Ly6G antibody-treated mice compared with control IgG-treated mice. Treatment with DNase 1 starting at 7 days after stroke also attenuated BBB disruption (Fig. 5c, d), increased microvessels (Fig. 5g, h), and improved capillary perfusion (Fig. 5k, l) at 14 days.

**PAD4 regulates BBB permeability and neovascularization.** Peptidylarginine deiminase 4 (PAD4) is a histone-modifying enzyme that is critical for NET formation[30,31]. Quantitative PCR revealed a sustained 29.3-fold increase in PAD4 mRNA expression in the ischemic cortex at 3 days after stroke compared with sham-operated brains (Fig. 6a). Immunoblotting found a marked

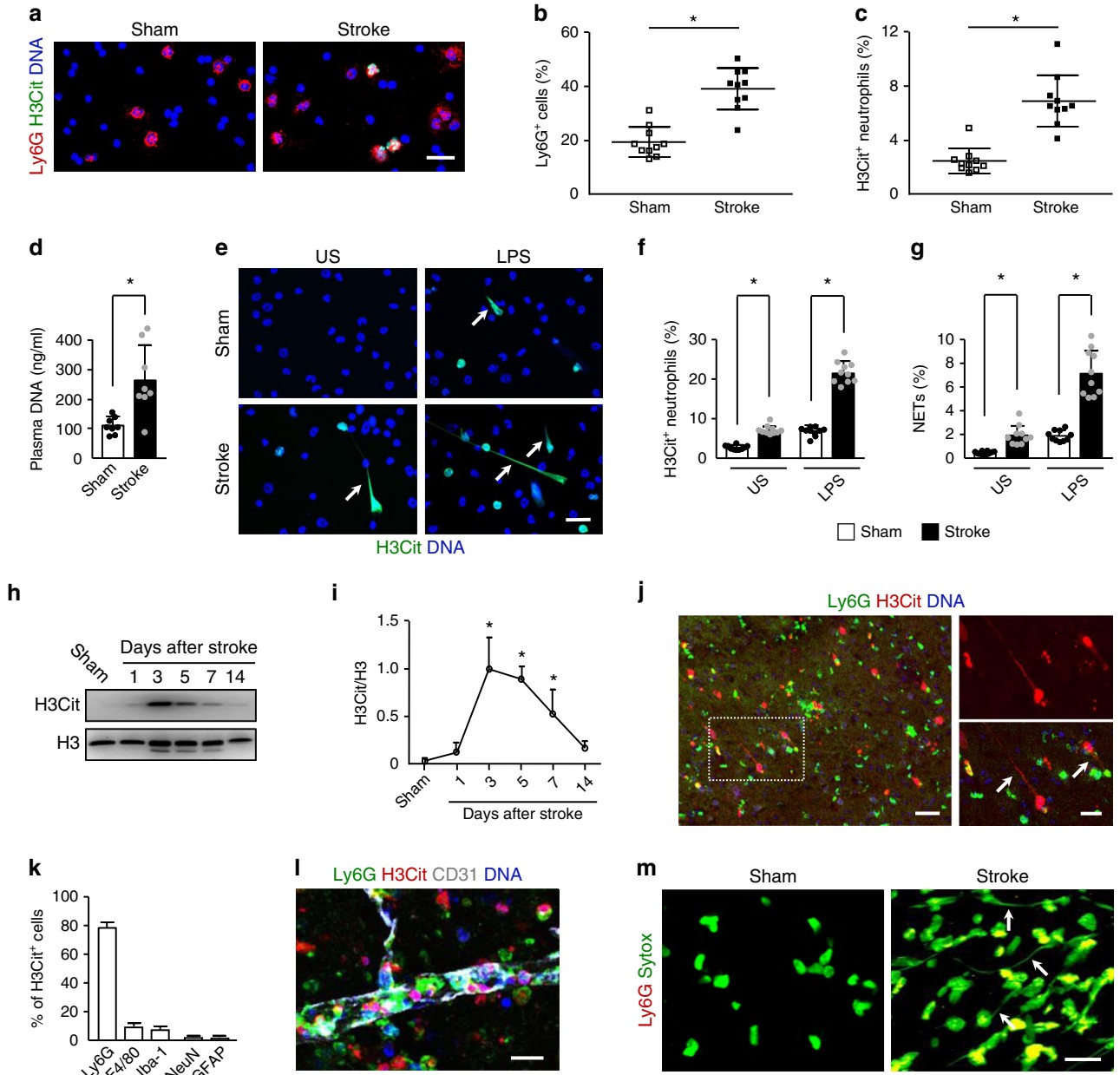

3.8-fold upregulation of PAD4 protein expression in these areas (Fig. 6b, c). To test the hypothesis that increased release of NETs, orchestrated by PAD4, participates in stroke recovery, we first studied the role of overexpression of PAD4 on vascular remodeling. Immunohistochemical analysis indicated extensive expression of recombinant adeno-PAD4-flag-infected cells in the cortex at 4 days after injection (Fig. 6d). Overexpression of PAD4 in the cortex was also confirmed by immunoblotting (Supplementary Fig. 5a, b). PAD4-flag was present in the neurons, neutrophils, macrophages/microglial cells, and microglial cells, but was rarely detected in astrocytes (Supplementary Fig. 5d-h). Injection of PAD4 adenovirus in ischemic mice produced ~2.6-fold more NETs than control virus (Fig. 6e and Supplementary Fig. 5c). We then observed a significant increase in the number of H3Cit+ neutrophils (Fig. 6f), whereas the number of H3Cit+ macrophages/microglial cells, H3Cit+ microglial cells, H3Cit+ neurons, and H3Cit+ astrocytes were not affect by PAD4 adenoviruses treatment (Supplementary Fig. 5i-l). At 14 days after stroke, multiphoton microscopy analysis of i.v.-injected FITC-

dextran showed a significant 2.5-fold increase in BBB permeability in the peri-infarct cortex in mice treated with PAD4 adenovirus (Fig. 6g, h). Administration of PAD4 adenovirus into ischemic mice also caused a significant reduction in CD13+ pericyte coverage on brain microvessels (Supplementary Fig. 5m, n). Moreover, vascular branches (Fig. 6i), microvascular length (Fig. 6j, k), and perfused cortical microvascular length (Fig. 6l, m) were significantly reduced in mice subjected to PAD4 adenovirus injection. The more severe BBB damage and poor neovascularization were functionally relevant. Compared with the control mice, PAD4-overexpressing mice had more severe neurological deficits as assayed by the beam walking test and forelimb force test (Fig. 6n-p).

To further establish the role of NETs in stroke recovery, we compared wild type (WT) with PAD4-deficient (PAD4−/−) mice or treated mice with the PAD inhibitor Cl-amidine. There was no significant difference in ischemic lesion between WT and PAD4−/− mice at 14 days (Supplementary Fig. 6a, b). Thus, this approach allowed us to provide evidence that effects of PAD4

**Fig. 3 Neutrophils form NETs presenting in the brain after stroke. a** Representative images of H3Cit (green) and Ly6G (red) double-positive cells in cytospins from sham-operated mice and ischemic mice at 3 days. DNA was visualized with Hoechst 33342 (blue). Bar = 30 μm. **b, c** Quantification of Ly6G-positive neutrophils in the total leukocyte population (**b**) and the percentage of H3Cit-positive neutrophils (**c**) in cytospins ($n = 10$ biologically independent experiments). Mann–Whitney test was applied with *$P < 0.0001$ (**b**), *$P < 0.0001$ (**c**). **d** Levels of plasma DNA were elevated at day 3 after stroke ($n = 6$ biologically independent experiments). Unpaired two-tailed Student's $t$-test was applied with *$P = 0.0026$. **e** Representative immunofluorescence images of isolated peripheral blood neutrophils from sham-operated mice and ischemic mice at 3 days. Neutrophils were incubated in the presence or absence of LPS for 2.5 h and stained with Hoechst 33342 (blue) and H3Cit (green). Arrows indicate NETs. US, unstimulated. Bar = 30 μm. **f, g** Quantification of the percentage of H3Cit-positive neutrophils (**f**) and NETs (**g**) in isolated neutrophils ($n = 10$ biologically independent experiments). One-way ANOVA test was applied with *$P < 0.0001$ (Sham vs. Stoke in US (**f**)), *$P < 0.0001$ (Sham vs. Stoke in LPS group (**f**)), *$P = 0.0235$ (Sham vs. Stoke in US (**g**)), *$P < 0.0001$ (Sham vs. Stoke in LPS group (**f**)). US, unstimulated. **h, i** Representative immunoblots of the time course of NETs appearance (**h**) and quantification of the H3Cit levels (**i**) in the peri-infarct cortex of mice subjected to stroke or sham operation ($n = 5$). One-way ANOVA test was applied with *$P < 0.0001$ (Sham vs. 3d), *$P < 0.0001$ (Sham vs. 5d), *$P = 0.0039$ (Sham vs. 7d). **j** Representative confocal images showing NET formation in the peri-infarct cortex of mice at 3 days after stroke. Inset is magnified on the right side. Arrows indicate NETs. Bar = 40 μm (left) and 20 μm (right). Arrows indicate extracellular DNA fibers. **k** Graphs compare the number of H3Cit⁺Ly6G⁺ neutrophils, H3Cit⁺F4/80⁺ macrophages/microglial cells, H3Cit⁺Iba1⁺ microglial cells, H3Cit⁺NeuN⁺ neurons, and H3Cit⁺ GFAP⁺ astrocytes in mice at 3 days after stroke ($n = 5$ biologically independent experiments). **l** Representative confocal image showing the formation of intravascular and intraparenchymal NETs at 3 days after stroke. Bar = 15 μm. Independent experiments are repeated at least three times. **m** Representative in-vivo multiphoton microscopy images of extracellular DNA (green) and neutrophils in the peri-infarct cortex of mice at 3 days. Extracellular DNA (green) were labeled with intravenous injection of Sytox green and neutrophils (red) with intravenous injection of PE-conjugated monoclonal Ly6G antibody. Arrows indicate extracellular DNA fibers. Bar = 20 μm. Independent experiments are repeated at least three times. Data are presented as mean ± SD. Source data underlying graph **b–d**, **f–i**, and **k** are provided as a Source Data file.

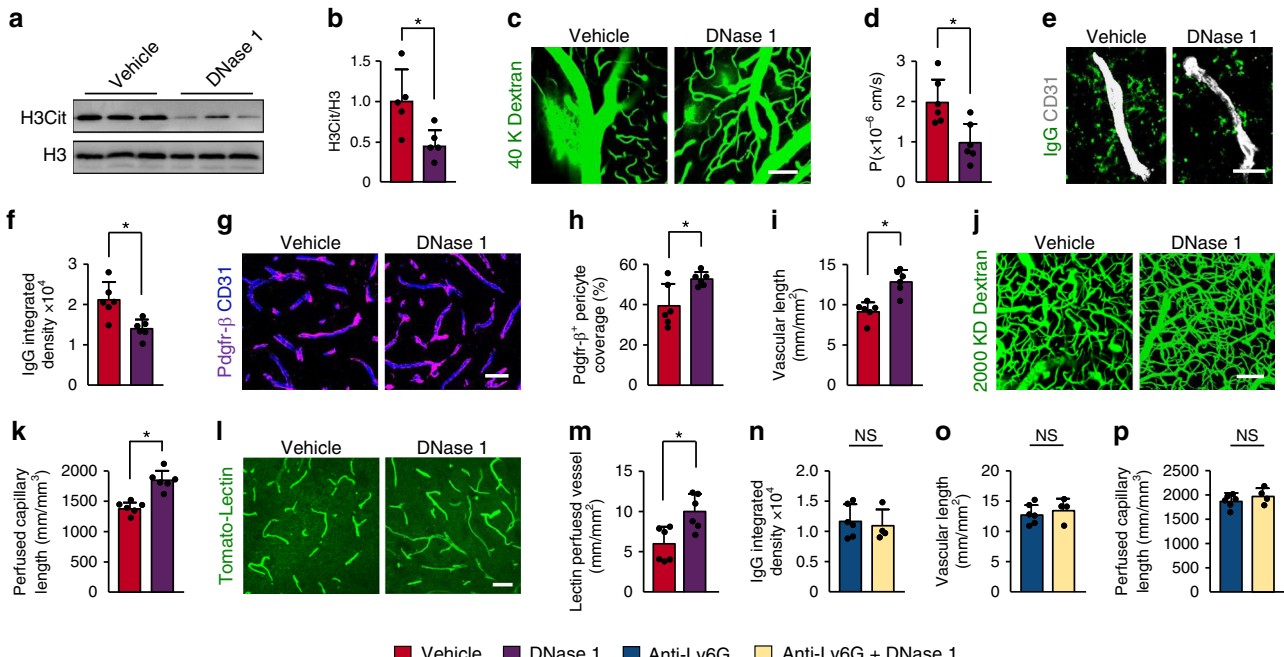

**Fig. 4 DNase 1 reduces BBB breakdown and increases neovascularization after stroke. a, b** Representative immunoblots (**a**) and quantification of H3Cit levels (**b**) in the peri-infarct cortex at 3 days in mice treated with vehicle or DNase 1 ($n = 5$), unpaired two-tailed Student's $t$-test was applied with *$P = 0.0228$. **c** Representative images of multiphoton microscopy of intravenously injected FITC-dextran (MW = 40,000 Da; green) leakage in cortical vessels at 14 days after stroke in mice treated with vehicle or DNase 1. Bar = 100 μm. **d** Quantification of the permeability (P) product of FITC-dextran for each group ($n = 6$ biologically independent animals), unpaired two-tailed Student's $t$-test was applied with *$P = 0.0083$. **e, f** Representative images of IgG deposits and CD31-positive microvessels (**e**) at 14 days after stroke in mice treated with vehicle or DNase 1, and quantification of extravascular IgG deposits (**f**) for each group ($n = 6$ biologically independent animals), unpaired two-tailed Student's $t$-test was applied with *$P = 0.0052$. Bar = 20 μm. **g, h** Representative images (**g**) and quantitative analysis (**h**) of Pdgfr-β-positive pericyte coverage on CD31-positive brain capillaries at 14 days ($n = 6$ biologically independent animals), unpaired two-tailed Student's $t$-test was applied with *$P = 0.0175$. Bar = 40 μm. **i** Quantification of microvascular length in the peri-infarct cortex at 14 days ($n = 6$ biologically independent animals), unpaired two-tailed Student's $t$-test was applied with *$P = 0.0007$. **j, k** In-vivo multiphoton microscopic images of perfused cortical capillaries with intravenously injected FITC-dextran (MW = 2000,000 Da) (**j**) and quantification of perfused capillary length (**k**) at 14 days ($n = 6$ biologically independent animals), unpaired two-tailed Student's $t$-test was applied with *$P = 0.0003$. Bar = 100 μm. **l, m** Representative images of tomato-lectin perfused vessels (**l**) and quantification of lectin perfused vessels (**m**) at 14 days ($n = 6$), Mann–Whitney test was applied with *$P = 0.0152$. Bar = 40 μm. **n–p** Quantification of extravascular IgG deposits (**n**), microvascular length (**o**), and perfused capillary length (**p**) in the peri-infarct cortex at 14 days ($n = 6$ for Anti-Ly6G, $n = 4$ for Anti-Ly6G + DNase 1), unpaired two-tailed Student's $t$-test was applied with $P = 0.6496$ (**n**), $P = 0.5222$ (**o**), $P = 0.2844$ (**p**). Data are presented as mean ± SD. Source data underlying graph **a**, **b**, **d**, **f**, **h**, **i**, **k**, and **m–p** are provided as a Source Data file.

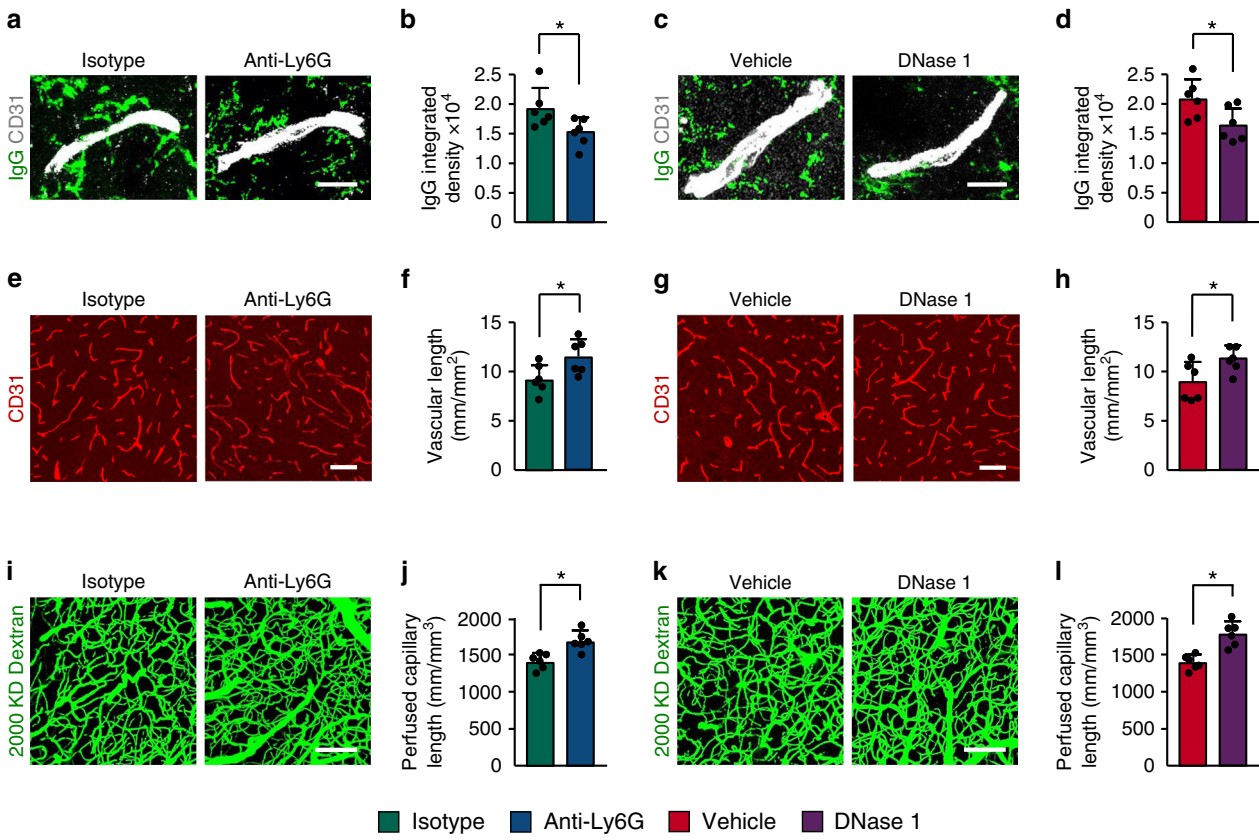

**Fig. 5 Increased vascular remodeling by delayed inhibition of NET formation. a–d** Representative confocal images (**a**, **c**) and quantitative analysis of IgG extravascular deposits (**b**, **d**) in the peri-infarct cortex at 14 days. Mice were subjected to stroke and treated with either anti-Ly6G antibody, control antibody, DNase 1, or vehicle starting at 7 days ($n = 6$), unpaired two-tailed Student's t-test was applied with *$P = 0.0392$ (**b**), *$P = 0.0384$ (**d**). Bar = 10 μm. **e–l** Representative confocal images (**e**, **g**) of CD31-positive microvessels and in-vivo multiphoton microscopy images of perfused cortical capillaries with intravenously injected FITC-dextran (**i**, **k**) in the peri-infarct cortex at 14 days in mice treated with either anti-Ly6G antibody, control antibody, DNase 1, or vehicle. Bar = 40 μm (**e**, **g**) and 100 μm (**i**, **k**). Quantification of microvascular density (**f**, **h**) and perfused capillary length (**j**, **l**) for each group ($n = 6$), unpaired two-tailed Student's t-test was applied with *$P = 0.00378$ (**f**), *$P = 0.0364$ (**h**), *$P = 0.0026$ (**j**), *$P = 0.0006$ (**l**). Data are presented as mean ± SD. Source data underlying graph **b**, **d**, **f**, **h**, **j**, and **l** are provided as a Source Data file.

deficiency on long-term outcomes are not secondary to the reduced lesion size. PAD4 deficiency did not alter the number of infiltrating neutrophils in the ischemic brain tissue (Supplementary Fig. 6c, d), whereas the levels of H3Cit in the lysates from the ischemic cortex at 3 days were greatly reduced in PAD4$^{-/-}$ mice and mice receiving Cl-amidine (Fig. 7a and Supplementary Fig. 7a). Similarly, examination of the brain tissues revealed that PAD4 deficiency or Cl-amidine reduced Sytox green-positive extracellular DNA fibers (Fig. 7b, c) in the ischemic cortex. Previous studies have shown that PAD4 may also function in other cells[32,33]. We found that PAD4 deficiency or Cl-amidine substantially reduced neutrophil NETs, as seen by the decrease in H3Cit$^+$ neutrophils (Fig. 7d, e). However, PAD4 deficiency did not significantly affect the number of H3Cit$^+$ macrophages/microglial cells, H3Cit$^+$ microglial cells, H3Cit$^+$ neurons, and H3Cit$^+$ astrocytes (Supplementary Fig. 6e-h). Mice deficient in PAD4 and mice treated with Cl-amidine exhibited a significant reduction in perivascular IgG deposits (Supplementary Fig. 7b, c) and vascular leakage of i.v.-injected FITC-dextran (Fig. 7f, g). IgG content in capillary-depleted brain tissues (Fig. 7h and Supplementary Fig. 8a) was also dramatically reduced in PAD4$^{-/-}$ mice and mice receiving Cl-amidine. Consistent with these findings, the tight-junction proteins ZO-1, claudin-5, and occludin, and the adherens junction protein VE-cadherin, which are required for BBB integrity[34], were enhanced in isolated brain microvessels in these mice (Fig. 7i and Supplementary Fig. 8b-e). Furthermore,

PAD4 deficiency or Cl-amidine treatment significantly increased vascular branches (Supplementary Fig. 7d, e), microvascular length (Fig. 7j and Supplementary Fig. 7f), the length of perfused microvessels (Fig. 7k), and tomato-lectin perfused vessels (Fig. 7l and Supplementary Fig. 7h) in the ischemic cortex. However, treatment with anti-Ly6G antibody or DNase 1 had no beneficial effect in PAD4$^{-/-}$ mice on BBB permeability (Fig. 7m and Supplementary Fig. 9a), microvascular length (Fig. 7n and Supplementary Fig. 9b), and perfused capillary length (Fig. 7o and Supplementary Fig. 9c). These data indicate that the effects of PAD4 deficiency on neovascularization and vascular remodeling are due to NETs. Furthermore, we found that PAD4 deficiency had no beneficial effect on BBB permeability ($1.18 × 10^4 ± 2702$ vs. $1.10 × 10^4 ± 2697$, $P = 0.66$, $n = 4$–6), microvascular length ($12.8 ± 1.68$ vs. $13.82 ± 2.31$, $P = 0.42$, $n = 4$–6), and perfused capillary length ($1871 ± 136$ vs. $1978 ± 91$, $P = 0.21$, $n = 4$–6) in mice subjected to anti-Ly6G antibody treatment. These findings suggest that PAD4 may primarily mediate the formation of neutrophil NETs after stroke. In parallel with the effects on vascular function, PAD4 deficiency or Cl-amidine treatment improved behavioral deficits in mice at 14 days after stroke (Fig. 7p-r).

### NETs are responsible for STING-mediated vascular remodeling.
The free DNA binds with cyclic GMP-AMP synthase to promote STING-dependent type I interferon (IFN) synthesis[35,36].

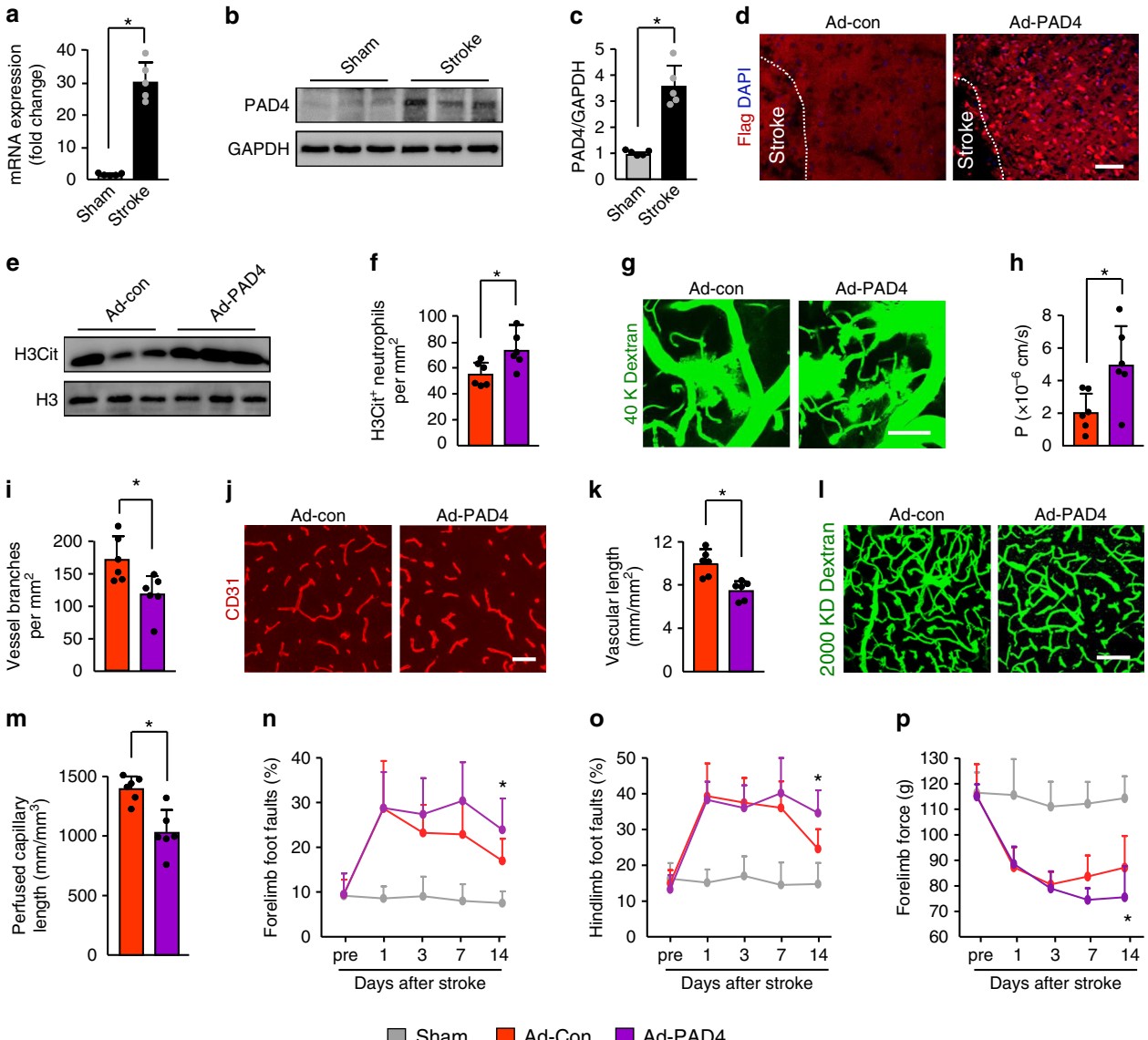

**Fig. 6 Overexpression of PAD4 exacerbates BBB breakdown and reduced revascularization. a** PAD4 mRNA expression was measured by quantitative real-time PCR in the peri-infarct cortex at 3 days, compared with sham-operated mice ($n = 5$); Mann–Whitney test was applied with *$P = 0.0079$. **b**, **c** Representative immunoblots (**b**) and quantification of PAD4 levels (**c**) in the peri-infarct cortex at 4 days after injection of control (Ad-Con) or PAD4 adenovirus (Ad-PAD4) ($n = 5$); Mann–Whitney test was applied with *$P = 0.0079$. **d** Representative images of recombinant Adeno-PAD4-flag-infected cells in the ischemic cortex. Mice were treated with recombinant Adeno-PAD4-flag (right panel) or empty adenovirus (left panel). At 4 days after injection, brain sections were stained with an antibody against Flag. Bar = 40 μm. **e** Representative immunoblots of H3Cit levels in the peri-infarct cortex at 3 days. **f** Quantification of the numbers of H3Cit-positive neutrophils in the peri-infarct cortex at 4 days after injection of control or PAD4 adenovirus ($n = 6$); unpaired two-tailed Student's $t$-test was applied with *$P = 0.0230$. **g** In-vivo multiphoton microscopic images of intravenously injected FITC-dextran (MW = 40,000 Da; green) leakage in cortical vessels at 14 days. Bar = 100 μm. **h** Quantification of the permeability (P) product of FITC-dextran for each group ($n = 6$); unpaired two-tailed Student's $t$-test was applied with *$P = 0.0244$. **i** Quantification of vascular branches in mice treated with control or PAD4 adenovirus at 14 days after stroke ($n = 6$), unpaired two-tailed Student's $t$-test was applied with *$P = 0.0170$. **j**, **k** Confocal images of CD31-positive microvessels (**j**) and quantification of microvascular density (**k**) in the peri-infarct cortex at 14 days ($n = 6$), unpaired two-tailed Student's $t$-test was applied with *$P = 0.0019$. Bar = 50 μm. **l**, **m** In-vivo multiphoton microscopic images of perfused cortical capillaries with intravenously injected FITC-dextran (**l**) and quantification of perfused capillary length (**m**) at 14 days after stroke ($n = 6$), unpaired two-tailed Student's $t$-test was applied with *$P = 0.0017$. Bar = 100 μm. **n–p** Overexpression of PAD4 worsened neurological outcomes in beam walking test (**n**, **o**) and forelimb force test (**p**) ($n = 10$). One-way ANOVA test was applied with *$P = 0.001$ (**n**), *$P = 0.0015$ (**o**), *$P < 0.0001$ (**p**) (Sham and Ad-Con). *$P = 0.0015$ (**n**), *$P = 0.0015$ (**o**), *$P = 0.0279$ (**p**) (Ad-Con and. Ad-PAD4). Data are presented as mean ± SD. Source data underlying graph **a–c**, **e**, **f**, **h**, **i**, **k**, **m–p** are provided as a Source Data file.

We found a marked tenfold increase in the levels of IFN-β in the cortical areas at 3 days after stroke (Fig. 8a), suggesting activation of the STING pathway. We then detected significantly increased levels of STING (Fig. 8b and Supplementary Fig. 10a) and robust induction of phosphorylated TANK-binding kinase 1 (pTBK1) and TBK1-dependent IFN regulatory factor 3 (IRF3) activation (Fig. 8b and Supplementary Fig. 10b, c). Treatment with the PAD inhibitor Cl-amidine reduced the levels of stroke-induced STING-mediated signaling in the cortical areas compared with vehicle-treated mice (Fig. 8c, d and Supplementary Fig. 10d–f).

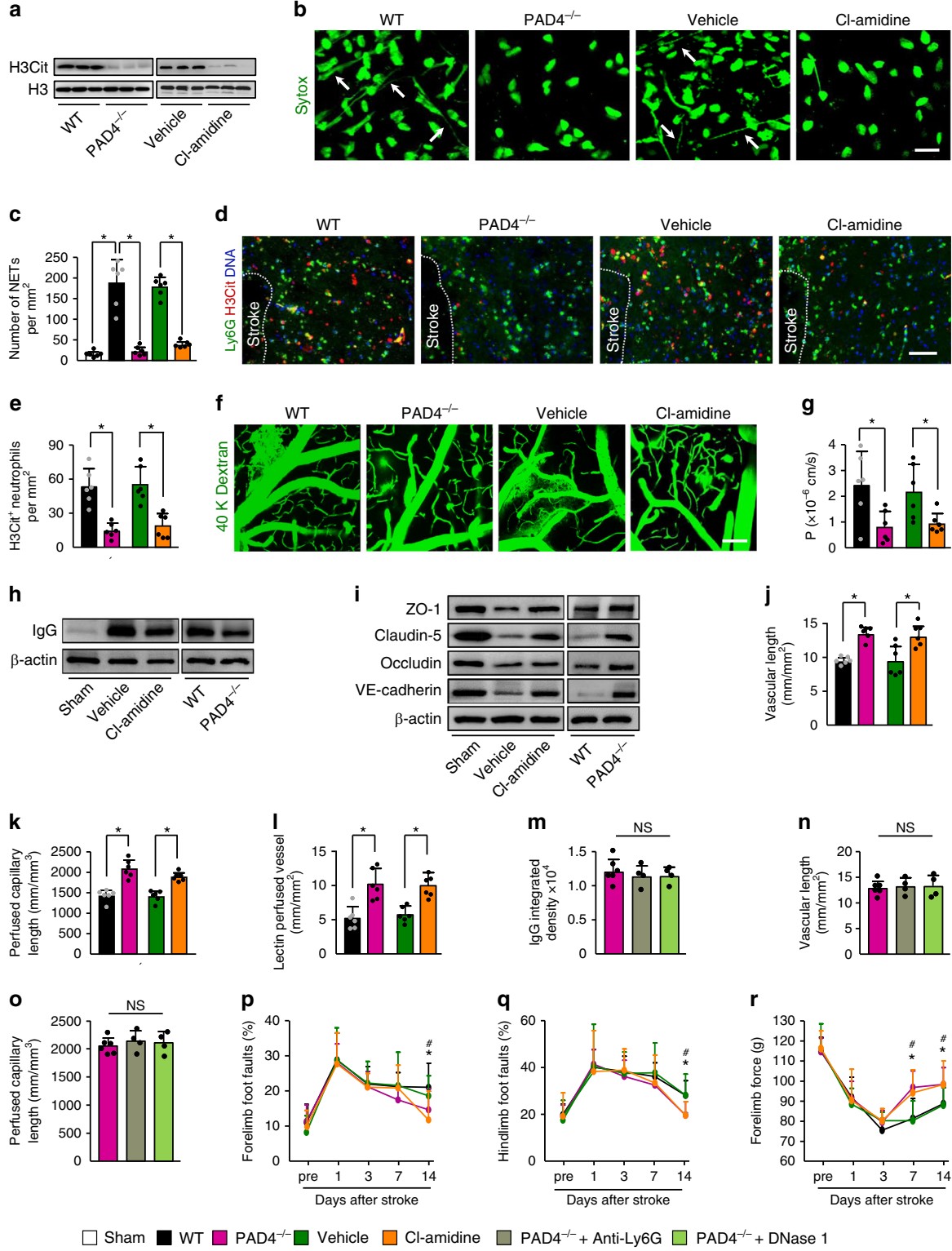

Neutrophil depletion also resulted in a significant reduction in the levels of STING and the STING downstream signaling molecules including pTBK1, pIRF3, and IFN-β in the ischemic brain relative to IgG-treated controls (Fig. 8e, f and Supplementary Fig. 10g-i). To test whether the signaling part mediated by STING is related to neutrophils, we isolated neutrophils from bone marrow of ischemic mice and stimulated them with LPS in the presence or absence of the PAD inhibitor Cl-amidine. We found that the STING-mediated signals were activated by LPS stimulation and

this effect was prevented by the addition of Cl-amidine (Fig. 8g, h and Supplementary Fig. 11a-c). Next, we investigated whether the IFN response affected the BBB permeability and vascular remodeling. Immunofluorescent staining and multiphoton microscopy indicated increased microvascular length (Fig. 8i, j) and perfused microvascular length (Fig. 8k, l) in the peri-infarct cortex in IFN receptor (IFNAR)-neutralizing antibody-treated mice as compared with control IgG-treated mice at day 14 after stroke. when analysing the i.v.-injected FITC-dextran, we observed a significant

**Fig. 7 PAD4 deficiency or pharmacologic inhibition promotes vascular remodeling after stroke. a** Representative immunoblots of H3Cit levels in the peri-infarct cortex at 3 days in WT and PAD4$^{-/-}$ mice, and WT mice treated with vehicle or the PAD inhibitor Cl-amidine. Independent experiments are repeated at least three times. **b** In-vivo multiphoton microscopic images of extracellular DNA (Sytox, green) in the peri-infarct cortex of mice at 3 days after stroke. Arrows indicate extracellular DNA fibers. Bar = 20 μm. Independent experiments are repeated at least three times. **c** Quantification of NETs for each group ($n = 6$). One-way ANOVA test was applied with *$P < 0.0001$ (Same vs. WT stroke), *$P < 0.0001$ (WT vs. PAD4$^{-/-}$), *$P < 0.0001$ (Vehicle vs. Cl-amidine). **d** Representative confocal images of neutrophil (Ly6G, green) and H3Cit (red) immunostaining in the peri-infarct cortex at 3 days. DNA was stained with Hoechst 33342 (blue). Bar = 40 μm. **e** Quantification of the numbers of H3Cit-positive neutrophils in the peri-infarct cortex at 3 days in WT and PAD4$^{-/-}$ mice, and WT mice treated with vehicle or the PAD inhibitor Cl-amidine ($n = 6$), unpaired two-tailed Student's $t$-test was applied with *$P = 0.0002$ (WT vs. PAD4$^{-/-}$), *$P = 0.0010$ (Vehicle vs. Cl-amidine). **f, g** In-vivo multiphoton microscopic images (**f**) of intravenously injected FITC-dextran leakage in cortical vessels at 14 days and quantification of the permeability (P) product of FITC-dextran (**g**) for each group ($n = 6$), unpaired two-tailed Student's $t$-test was applied with *$P = 0.0197$ (WT vs. PAD4$^{-/-}$), *$P = 0.0260$ (Vehicle vs. Cl-amidine). Bar = 100 μm. **h** Representative immunoblots of IgG levels in capillary-depletion brain tissue at 14 days in WT and PAD4$^{-/-}$ mice, and WT mice treated with vehicle or Cl-amidine. Independent experiments are repeated at least three times. **i** Representative immunoblots of the tight-junction protein ZO-1, claudin-5, and occludin, and the adherens junction protein VE-cadherin in isolated brain microvessels at 14 days. Independent experiments are repeated at least three times. **j–l** Quantification of microvascular density (**j**), perfused capillary length (**k**), and tomato-lectin perfused vessels (**l**) in the peri-infarct cortex at 14 days ($n = 6$). Unpaired two-tailed Student's $t$-test was applied with *$P < 0.0001$ (WT vs. PAD4$^{-/-}$ (**j**)), *$P = 0.0093$ (Vehicle vs. Cl-amidine (**j**)), *$P = 0.0001$ (WT vs. PAD4$^{-/-}$ (**k**)), *$P < 0.0001$ (Vehicle vs. Cl-amidine (**k**)) *$P = 0.0015$ (WT vs PAD4$^{-/-}$ (**l**)), *$P = 0.0013$ (Vehicle vs. Cl-amidine (**l**)). **m–o** Quantification of extravascular IgG deposits (**m**), microvascular length (**n**), and perfused capillary length (**o**) in the peri-infarct cortex at 14 days after stroke ($n = 6$ for PAD4$^{-/-}$, $n = 4$ for PAD4$^{-/-}$ + Anti-Ly6G and PAD4$^{-/-}$ + DNase 1). One-way ANOVA test was applied with $P = 0.7263$ (**m**), $P = 0.9547$ (**n**), $P = 0.7304$ (**o**). **p–r** PAD4 deficiency or Cl-amidine treatment improved neurological functions in beam walking test (**p, q**) and forelimb force test (**r**) ($n = 10$). One-way ANOVA test was applied with *$P = 0.0407$ (**p**), *$P = 0.0056$ (**q**), *$P = 0.0013$ (7d (**r**)) and *$P = 0.0172$ (14d (**r**)) (PAD4$^{-/-}$ and WT), #$P = 0.0439$ (**p**) and #$P = 0.0210$ (**q**), #$P = 0.0091$ (7d (**r**)) and #$P = 0.0394$ (14d (**r**)) (Cl-amidine and vehicle). Data are presented as mean ± SD. Source data underlying graph **a**, **c**, **e**, and **g–r** are provided as a Source Data file.

reduction in BBB damage in mice treated with IFNAR-neutralizing antibody (Fig. 8m, n). Similarly, STING silencing by adenoviral short hairpin RNA (shRNA) administration into the brain of mice resulted in a significant increase in the length of microvessels (Fig. 8i, j) and perfused cortical microvessels (Fig. 8k, l), and a substantial reduction in BBB disruption (Fig. 8m, n). Together, these findings suggest that NETs induce the activation of the STING pathway, and that inhibition of the type I IFN response can enhance neovascularization and vascular repair.

## Discussion

Neovascularization and perfusion of the vascular structure in the peri-ischemic brain have important roles in stroke recovery[37,38]. However, these newly formed vessels are permeable and not yet fully developed[28]. The opened BBB leads to increased extravasation of immune cells and blood-derived toxic proteins[39,40]. Therefore, stability of the blood vessels and restoration of the damaged BBB may be crucial to maintaining a stable brain microenvironment. However, the mechanisms underlying vascular plasticity and the potential link between BBB opening and neurovascular dysfunction after stroke are not fully understood.

Our data showed the accumulation of neutrophils in the brain during all stages of stroke, suggesting that neutrophils may cause delayed vascular damage. Indeed, neutrophil depletion reduced BBB breakdown at 14 days after stroke, demonstrating a role for neutrophils in the induction of vascular impairment during the later phases. We then found neutrophil depletion increased neovascularization and vascular perfusion. These results suggest that neutrophils are critical to disrupt stroke-induced new vessel formation and stabilization. These findings are also supported by previous studies showing that chronic BBB breakdown is associated with microvascular reductions[34]. Interestingly, in Alzheimer's disease models, depletion of neutrophils by 300 μg of anti-Ly6G antibody every second day for 1 month reduced Alzheimer's disease pathogenesis and improved memory[41]. However, anti-integrin therapies were previously shown to induce progressive multifocal leukoencephalopathy in patients with autoimmune disorders[42]. Further investigation of whether this long

period of neutrophil depletion can induce side effects will be essential.

NETs release many cytotoxic proteases such as histone, elastase, and MPO, which directly induce endothelial cell damage to increase vascular permeability[43]. We observed that neutrophils isolated from ischemic mice formed more spontaneous NETs and showed a greater tendency to make NETs after exposure to LPS. Consistently, our data showed that stroke activated neutrophils to release excessive NETs within the vasculature and the parenchyma, concordant with elevated circulating DNA. Digestion of NETs with DNase 1 significantly reduced BBB damage, which was accompanied by increased pericyte coverage on microvessels and formation of new functional vessels, supporting NET formation as a cause of vascular injury[44,45]. These findings are in agreement with a previous report in mice with acute ischemic stroke[46] and also strongly suggest that NETs may be critical for neutrophil-dependent vascular destabilization and regression during the delayed stages after stroke.

PAD4 is a key enzyme in chromatin decondensation[30,31]. We found that PAD4 was markedly upregulated in the peri-ischemic cortex. Overexpression of PAD4 resulted in amplified vascular damage and reduced neovascularization by releasing more NETs. Consistently, we showed that inhibition of NET formation by PAD4 genetic knockout or pharmacological inhibitor reduced loss of cerebrovascular integrity, increased neovascularization and capillary perfusion, and improved functional recovery. These results demonstrated that, by increasing NET formation, PAD4 impaired delayed vascular remodeling after stroke.

Our data revealed that stroke led to upregulation of the DNA sensor STING, activation of TBK1 and IRF3, and induction of the IRF3-dependent IFN-β synthesis. Silencing STING or administration of blocking antibody to IFNAR in mice increased vascular regeneration and repair. Furthermore, we found that the PAD inhibitor Cl-amidine suppressed the activation of the STING pathway and the production of IFN-β. Thus, the STING-mediated type I IFN response may link NETs and ischemic vascular remodeling.

In summary, our findings demonstrated that stroke caused neutrophil accumulation in the brain, releasing toxic signals such as NETs, which promoted subsequent activation of STING-dependent

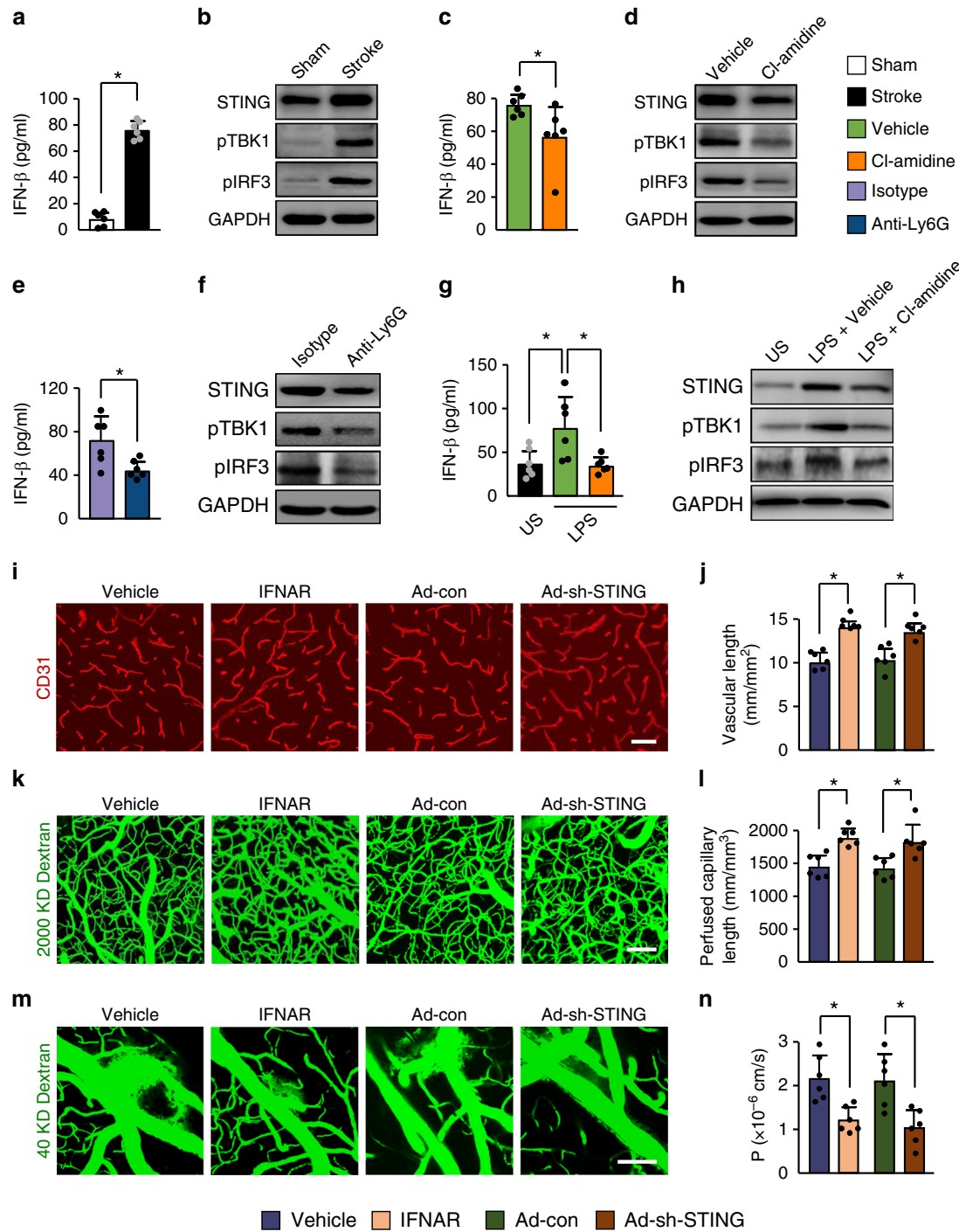

type I IFN-β production. Importantly, we also showed that increased NET formation or impaired NET clearance was detrimental to revascularization and vascular repair after stroke. We suggest that NETs are key targets for promoting stroke-mediated neovascularization and the resulting functional recovery.

## Methods

**Animal stroke model.** Animal protocols were reviewed and approved by the Animal Care and Use Committee of the Institutes of Brain Science of Fudan University and were conducted in accordance with the ethical regulations. PAD4$^{-/-}$ mice on a C57BL/6J background were purchased from The Jackson Laboratory. Age-matched WT C57BL/6 mice (SLAC Laboratory Animal Co. Ltd, Shanghai, China) were used as controls. All mice were housed in a temperature-controlled environment (22 ± 2 °C) on a 12 h light–dark cycle with food and water available ad libitum. Room humidity was controlled at 55 ± 5%. Focal cortical cerebral ischemia

was induced by electrocoagulation of the distal portion of the right middle cerebral artery (MCA)[28,47]. Male mice weighing 23–26 g were anesthetized with 1–1.5% isoflurane in 30% oxygen and 70% nitrous oxide. After making a 2 cm curved skin incision between the right eye and the right ear using surgical scissors, the temporal muscle was retracted laterally. Under an operating microscope, a 1.5 mm-diameter window was opened using a high-speed micro drill (Stoelting, CellPoint Scientific, Maryland) just rostral to the foramen ovale. The dura mater was carefully opened and the distal MCA was exposed and isolated, electrocauterized using bipolar electrocoagulation forceps, and disconnected just distal to crossing the olfactory tract. Immediately following the occlusion, the right common carotid artery (CCA) was occluded with a microvascular clip (Fine Science Tools, Foster City, CA) for 15 min. After closing of the surgical wounds, mice were allowed to recover from the anesthesia. Sham-operated animals only had the MCA and the CCA exposed.

For cerebral infarction, six coronal sections (20 μm) were stained with hematoxylin and eosin. Infarct volume was measured using the NIH ImageJ 1.46r software[47] and was presented as percentage of the contralateral hemisphere.

**Fig. 8 STING-mediated effects on vascular remodeling are due to NETs. a** Levels of IFN-β were increased in the ischemic cortex at 3 days, compared with sham-operated brains ($n = 6$), unpaired two-tailed Student's $t$-test was applied with *$P < 0.0001$. **b** Immunoblot analysis of STING, phosphorylated TBK1 (pTBK1), and pIRF3 in the cortex of mice without stroke or at day 3 after stroke. **c** Levels of IFN-β in the ischemic cortex at 3 days in mice treated with control or the PAD inhibitor Cl-amidine ($n = 6$). Mann–Whitney test was applied with *$P = 0.0152$. **d** Immunoblot analysis of STING, pTBK1, and pIRF3 in the ischemic cortex at day 3. **e** Levels of IFN-β in the ischemic cortex at 3 days in mice treated with control antibody or anti-Ly6G antibody ($n = 6$), unpaired two-tailed Student's $t$-test was applied with *$P = 0.0148$. **f** Immunoblot analysis of STING, pTBK1, and pIRF3 in the ischemic cortex at day 3. **g** Levels of IFN-β in isolated bone marrow neutrophils from ischemic mice. Neutrophils were stimulated with LPS in the presence or absence of the PAD inhibitor Cl-amidine ($n = 6$). One-way ANOVA test was applied with *$P = 0.0230$ (Stoke vs. Vehicle), *$P = 0.0160$ (Vehicle vs. Cl-amidine). US, unstimulated. **h** Immunoblot analysis of STING, pTBK1, and pIRF3 in isolated neutrophils for each group ($n = 5$). **i, j** Confocal images of CD31-positive microvessels (**i**) and quantification of microvascular density (**j**) in the peri-infarct cortex at 14 days in mice treated with control IgG or IFNAR-neutralizing antibody, and STING shRNA or control adenovirus ($n = 6$), unpaired two-tailed Student's $t$-test was applied with *$P < 0.0001$ (Vehicle vs. IFNAR), *$P = 0.0007$ (Ad-con vs. Ad-sh-STING). Bar = 40 μm. **k, l** In-vivo multiphoton microscopic images of perfused cortical capillaries with intravenously injected FITC-dextran (**k**) and quantification of perfused capillary length (**l**) at 14 days after stroke ($n = 6$), unpaired two-tailed Student's $t$-test was applied with *$P = 0.0006$ (Vehicle vs. IFNAR), *$P = 0.0080$ (Ad-con vs. Ad-sh-STING). Bar = 100 μm. **m, n** In-vivo multiphoton microscopic images (**m**) of intravenously injected FITC-dextran leakage in cortical vessels at 14 days and quantification of the permeability (P) product of FITC-dextran (**n**) for each group ($n = 6$), unpaired two-tailed Student's $t$-test was applied with *$P = 0.0028$ (Vehicle vs. IFNAR), *$P = 0.0053$ (Ad-con vs. Ad-sh-STING). Bar = 100 μm. Data are presented as mean ± SD. Source data underlying graph **a–h**, **j**, **l**, and **n** are provided as a Source Data file.

**In-vivo multiphoton microscopy**. Mice were anesthetized with 1–1.5% isoflurane in 30% oxygen and 70% nitrous oxide, and kept on a heating plate (37 ± 0.5 °C). After fixation in a custom-made head holder, a craniotomy was created over the right somatosensory cortex (centered 2.5 mm lateral and 1.5 mm posterior to the bregma) using a high-speed micro drill. A sterile cover glass was placed above the window and fixed with dental cement. Mice were imaged through cranial windows using an upright multiphoton laser-scanning microscope (FluoView FVMPE-RS, Olympus, Japan) with an Olympus XL Plan N ×25/1.05 WMP ∞ / 0–0.23/FN/18 dipping objective. Two-photon excitation was accomplished with MAITAI eHPDS-OL and Spectra Physics InSight DS-OL lasers (Mai Tai, Spectra-Physics, Santa Clara, CA). Emitted fluorescence was detected through a 495–540 nm bandpass filter.

To analyze microvascular perfusion, a 0.1 mL bolus of 10 mg/mL FITC-dextran (2,000,000 Da, Sigma-Aldrich; St. Louis, MO) was injected i.v.[28,48,49]. Z-stack images at 5 μm steps were acquired from 200 to 500 μm below the surface of the cortex. The area scanned was 500 × 500 μm with 1024 × 1024 pixel resolution. Images were reconstructed by Olympus FV 10-ASW software. The total length of microvessels <6 μm in diameter was determined using NIH ImageJ 1.46r software. Data are presented as mm of vessels per mm³ of cortical tissue.

Cerebrovascular permeability was evaluated by time-lapse imaging acquired every 3 min for 30 min after i.v. injection of FITC-dextran (40,000 Da, Sigma-Aldrich, 0.1 mL of 10 mg/mL). The fluorescence of randomly chosen 20 × 20 μm² regions of interest within the vessel and corresponding areas within the perivascular brain parenchyma were recorded[28,50].

To detect neutrophils, phycoerythrin (PE)-conjugated monoclonal Ly6G antibody (1A8 clone; 3 μg, 12-9668-80, eBioscience) was injected i.v. into mice[51]. Blood vessels were visualized by i.v. injection of FITC-dextran (2,000,000 Da, Sigma-Aldrich, 0.1 mL of 10 mg/mL). Time-lapse images were acquired from 200 to 300 μm below the cortical surface every 3 min for 30 min.

To visualize NETs, mice were injected with 5 μL i.v. Sytox Green (S7020, Invitrogen)[52] 30 min before imaging. Z-stacks of four fields were taken and Sytox Green-positive single NET fibers were counted using NIH ImageJ[53]. Data are presented as the average number of fibers per mm² of cortical tissue. Sytox Green-positive intact cells were excluded from the quantification[53].

**Neutrophil depletion**. Mice received an intraperitoneal (i.p.) injection of 100 μg monoclonal anti-mouse Ly6G (1A8 clone; specific for neutrophils, BE0075-1, BioXCell, NH) 24 h or 7 days after cerebral ischemia[44]. The mice were then injected every second day for 3, 7, or 14 days until killing. Rat IgG2a isotype control was administered in the same way. The dose of anti-Ly6G antibody and the length of treatment were chosen based on previous studies[54]. At 14 days after stroke, no significant differences were observed in body weight (24.01 ± 0.91 g vs. 24.05 ± 0.70 g; $P = 0.90$, $n = 12$) and survival rate (91.7% vs. 91.7%) between neutrophil depletion and control groups. To verify neutrophil depletion, blood neutrophil levels were evaluated by flow cytometry (BD LSRFortessaTM, BD Biosciences) using a FITC-conjugated rat monoclonal anti-mouse neutrophil antibody (anti-7/4; ab53453, Abcam) and analyzed using FlowJo V10 software (Tree Star, Inc., Ashland, OR).

**DNase 1 treatment**. Mice were injected with 10 μg i.v. and 50 μg i.p. DNase 1 (Deoxyribonuclease 1 human recombinant; enz-319-10000IU, ProSpec, Israel) 24 h or 7 days after cerebral ischemia and then 50 μg i.p. every 12 h until killing on day 3 or day 14[55]. Control mice were injected with vehicle (8.77 mg/mL sodium chloride and 0.15 mg/mL calcium chloride).

**PAD inhibitor**. Stock solution of PAD inhibitor Cl-amidine (506282, Millipore) was dissolved in dimethyl sulfoxide (DMSO) (Sigma-Aldrich). The stock solution was dissolved in saline (5% v/v) and injected i.p. at 10 mg/kg 24 h after cerebral ischemia and then every day until the mice were killed[56]. Vehicle (saline containing 5% DMSO) was administered in the same way.

**Type I IFNAR-neutralizing antibody**. Mice were injected i.p. with 10 mg/kg IFNAR-neutralizing antibody (MAR1-5A3; BE0241, BioXCell, NH) 24 h after cerebral ischemia and then every second day until killing on day 14[57]. Control mice were injected with mouse IgG isotype control (MOPC-21; BE0083, BioXCell).

**Injection of adenoviruses**. Recombinant PAD4 adenovirus (Adeno-PAD4; Adeno-CMV-Padi4-3*flag-tagged, $1.26 \times 10^{11}$ plaque-forming-unit/mL) and empty adenovirus (Adeno-CMV-3*flag-tagged) were produced by Hanbio Biotechnology (Shanghai, China). Adenoviral vector expressing STING shRNA (pDKD-CMV-Puro-U6-Tmem173 (STING)-shRNA, $1.0 \times 10^{10}$ plaque-forming-unit/mL) and control shRNA (pDKD-CMV-Puro-U6-shRNA) were produced by Obio Technology (Shanghai, China). Both of them (2 μL) were stereotactically injected into the right cortex (coordinates: 0.2 mm posterior to bregma, 2.0 mm lateral to midline, and 1.0 mm ventral to skull surface)[58] of the brain 24 h before cerebral ischemia and analyzed 4 days after injection.

**Blood counts**. At 14 days after cerebral ischemia, mice were bled from the retro-orbital plexus under isoflurane anesthesia. Blood was collected into a EDTA-containing tube. The samples were analyzed using a SYSME X Hematology Analyzer (Sysme, Japan).

**Behavior**. Behavioral tests were performed before and 1, 3, 7, and 14 days after ischemia by an investigator blinded to the experimental groups. A grip strength meter (Bio-Seb, Vitrolles, France) was used to assess the peak force exerted by an animal when the animal released the forepaws from a grid[59]. A digital reading of six trials was conducted for each mouse and the average force was used for analysis. For the beam walking test[28,60], the mice were put on one end of a wooden beam (12 mm in diameter, 1.2 m long, and 45 cm high), and the total numbers of limb steps and the numbers of foot faults were recorded. The percentage of foot faults to total steps that occurred within 10 min was calculated. A foot fault was defined as any paw slips off the top surface of the beam. Before surgery, mice were trained for 3 days.

**Quantification of plasma DNA**. Plasma was collected from whole blood by centrifugation at $150 \times g$ for 15 min. DNA in plasma was quantified according to the manufacturer's instructions using the Quant-iT PicoGreen dsDNA Assay kit (Invitrogen).

**IFN-β measurement**. Ischemic cortical tissues were homogenized in RIPA lysis buffer (Millipore) including protease inhibitor cocktails (Roche Diagnostics GmbH, Mannheim, Germany). The levels of IFN-β was measured using the VeriKine™ Mouse Interferon Beta ELISA Kit (42400-1, PBL Assay Science, NJ) according to the manufacturer's instructions.

**Flow cytometry**. Peripheral blood was subjected to red blood cell lysis buffer (155 mmol/L NH₄Cl, 10 mmol/L KHCO₃, and 0.1 mmol/L Na₂EDTA). Cells were washed with phosphate-buffered saline (PBS) containing 1% bovine serum albumin

(BSA) and resuspended in rat anti-mouse CD16/32 Fc block (2.4G2 clone; 1 : 100, 553141, BD Pharmingen) in PBS. Cell suspension was incubated with Allophycocyanin-cyanine dye (APC-Cy7)-conjugated antibody to CD11b (Integrin alpha M, M1/70 clone; 1 : 200, 557657, BD Pharmingen), V450-conjugated antibody to CD45 (30-F11 clone; 1 : 200, 560501, BD Pharmingen) and PE-conjugated antibody to Ly6G (1A8 clone; 1 : 200, eBioscience). Flow cytometry was performed on a BD LSRFortessa™ (BD Biosciences) and data were analyzed with FlowJo V10 software (Tree Star, Inc., Ashland, OR).

**Cytospin NET analysis.** Whole blood collected from the retro-orbital sinus was lysed with red blood cell lysis buffer. The cells were resuspended in 7.5% BSA in PBS and plated on slides using a Shandon Cytospin 4 (Thermo Scientific). Slides were fixed in 4% paraformaldehyde at 4 °C overnight and incubated with rabbit anti-H3Cit (1 : 1000, ab5103, Abcam) and rat anti-Ly6G (1 : 200, 551459, BD Pharmingen) antibodies overnight at 4 °C, then incubated with Alexa Fluor 488-conjugated donkey anti-rabbit and Alexa Fluor 594-conjugated donkey anti-rat secondary antibodies (1 : 1000, Invitrogen). DNA was stained with Hoechst 33342 (1 : 10,000, Invitrogen). Images were obtained using an Olympus BX 63 microscope and an Olympus FV 1000 confocal microscope.

**Neutrophil isolation and in vitro NET assay.** Neutrophils were isolated from sham-operated and ischemic mice by Percoll (GE Healthcare) gradient centrifugation followed by hypotonic lysis of red blood cells[55]. Neutrophil purity was routinely >90% as assessed by Wright-Giemsa staining of cytospins. Freshly isolated neutrophils ($5 \times 10^5$ cells/mL) were suspended in RPMI-1640 (Gibco, MA) and seeded in 48-well glass-bottomed plates in a 5% $CO_2$ atmosphere at 37 °C for 30 min before stimulation. Following incubation with 10 μg/mL *Klebsiella pneumoniae* LPS (Sigma) at 37 °C for 2.5 h, cells were fixed in 2% paraformaldehyde in PBS. Cells were blocked with 3% BSA in PBS and incubated with rabbit anti-H3Cit antibody (1 : 1000, ab5103, Abcam) overnight at 4 °C. After three washes, Alexa Fluor 488 donkey anti-rabbit IgG was added for 30 min at room temperature. DNA was stained with Hoechst 33342 (1 : 10,000, Invitrogen). In some experiments, neutrophils from bone marrow were incubated with LPS in the presence of the PAD inhibitor Cl-amidine (200 μM) or vehicle (saline containing 0.5% DMSO)[31]. After the incubation, cells were lysed and protein samples were prepared for western blot analysis.

**MPO activity assay.** Ipsilateral brain cortex was homogenized in 50 mM potassium phosphate buffer, centrifuged, and suspended in 0.5% cetyltrimethylammonium bromide (Sigma-Aldrich) in potassium phosphate buffer. The suspensions were sonicated for 30 s with three freeze–thaw cycles in liquid nitrogen. After centrifugation, 40 μL of supernatant was incubated with 100 μL tetramethylbenzidine solution (Sigma-Aldrich) and the reaction was stopped with 100 μL 2 N HCl. The optical density was measured at 450 nm (Bio-Tek, Vermont)[60]. MPO activity was expressed in equivalent units by comparison with a reference curve generated using purified MPO (Sigma-Aldrich).

**Preparation of brain microvessels and capillary-depleted brain homogenates.** Brains were removed and the meninges and large surface vessels were discarded. Brain tissue was homogenized in 16% dextran (Sigma-Aldrich) in PBS containing 2% fetal bovine serum, centrifuged at $6000 \times g$ for 15 min. The supernatant was collected and centrifuged again to obtain capillary-depleted brain homogenates. Pellets were resuspended in PBS containing 1% BSA and passed through a 100 and 45 μm cell strainer (BD Falcon, CA). Microvessels were trapped on top of the 45 μm strainer.

**Western blotting.** Brain tissues, microvessels, and capillary-depleted brain homogenates were lysed in RIPA lysis buffer (Millipore) containing protease inhibitor cocktails (Roche Diagnostics, Mannheim, Germany). Equal amounts of protein were loaded on SDS-polyacrylamide gel electrophoresis gels and transferred onto polyvinylidene difluoride membranes. The primary antibodies used were as follows: rabbit anti-Histone H3 (anti-H3; 1 : 1000, 9715), rabbit anti-pTBK1 (1 : 1000, 5483), rabbit anti-pIRF3 (1 : 1000, 4947), rabbit anti-β-actin (1 : 2000, 4970), rabbit anti-reduced glyceraldehyde-phosphate dehydrogenase (GAPDH; 1 : 2000, 5174, all from Cell Signaling Technology, MA), rabbit anti-H3Cit (1 : 1000, ab5103), rabbit anti-CD144 (vascular endothelial cadherin, VE-cadherin; 1 : 1000, ab33168), rabbit anti-occludin (1 : 1000, ab167161), rabbit anti-claudin-5 (1 : 1000, ab15106, all from Abcam), rat anti-Ly6G (1 : 1000, 551459, BD Pharmingen), mouse anti-PAD4 (O94H5 clone; 1 : 1000, 684202, Biolegend), rabbit anti-Zonula occludens-1 (ZO-1; 1 : 1000, 617300, Invitrogen), sheep anti-stimulator of IFN genes (STING; 1 : 1000, AF6516, R&D Systems). The membranes were incubated overnight with primary antibodies, followed by incubation with horseradish peroxidase-conjugated anti-rabbit, anti-mouse, and anti-rat secondary antibodies. Blots were processed using Image Lab-5.2.1 software.

**Quantitative real-time PCR analysis.** Total RNA was extracted from cortical brain tissue using RNA Simple Total RNA Kit (TIANGEN Biotech Co. Ltd, China). The PCR was performed with equal amounts of cDNA in a Mastercycler ep realplex machine (Eppendorf, Germany). The relative gene expression levels were normalized to GAPDH.

**Primers.** Primers used for reverse-transcriptase PCR were as follows:
PAD4 forward 5′-TCTGCTCCTAAGGGCTACACA-3′
PAD4 reverse 5′-GTCCAGAGGCCATTTGGAGG-3′
GAPDH forward 5′-AATGTGTCCGTCGTGGATCTGA-3′
GAPDH reverse 5′-GATGCCT GCTTCACCACCTTCT-3′.

**Immunohistochemistry.** Immunohistochemical staining was conducted on 20 μm frozen coronal sections. The following primary antibodies were used: rat anti-Ly6G (1 : 200, 551459, BD Pharmingen), rat anti-CD31 (PECAM-1; 1 : 200, 550274, BD Pharmingen), goat anti-CD31 (1 : 200, AF3628, R&D Systems), rabbit anti-H3Cit (1 : 1000, ab5103, Abcam), FITC-conjugated rat anti-aminopeptidase N (CD13; 1 : 200, 558744, BD Pharmingen), goat anti-platelet-derived growth factor receptor-β (Pdgfr-β; 1:200, AF385, R&D Systems), rat anti-F4/80 (1 : 200, ab6640, Abcam), goat anti-Iba1 (1 : 200, ab5076, Abcam), goat anti-GFAP (1 : 200, ab53554, Abcam), mouse anti-neuronal nuclei (NeuN; 1 : 200, MAB377, Millipore), rabbit anti-NeuN (1 : 200, ab177487, Abcam), and mouse anti-Flag (1 : 200, #8146, Cell Signaling Technology, MA). The secondary antibodies used were as follows: Alexa Fluor 594-conjugated donkey anti-rabbit IgG (1 : 1000, A-21207), Alexa Fluor 594-conjugated donkey anti-goat IgG (1 : 1000, A-11058), Alexa Fluor 488-conjugated donkey anti-rat IgG (1 : 1000, A-21208), Alexa Fluor 488-conjugated donkey anti-mouse IgG (1 : 1000, A-21202), Alexa Fluor 647-conjugated donkey anti-goat IgG (1 : 1000, A-21447), and biotin-donkey anti-rat IgG (1 : 1000, A18743, all from Invitrogen, Waltham, MA). DNA was stained with Hoechst 33342 (1 : 10,000, H3570, Invitrogen). Staining was visualized by Olympus FV 1000 laser-scanning confocal microscope and an Olympus BX 63 microscope. For each animal, three fields from the peri-infarct cortex in each section were digitized under ×40 objective. Images were processed using Olympus FV 10-ASW 4.2 Viewer software and ImageJ 1.46r software. The length of CD31-positive vessels was measured using the ImageJ plugin "Analyze Skeleton" length analysis tool. The areas of CD13 or Pdgfr-β-immunostained vessels were measured using the ImageJ area measurement tool and were expressed as a percentage of the CD31-positive area in 0.42 mm$^2$ regions. The numbers of Ly6G-positive and H3Cit-positive cells in the traced area were determined.

**Statistical analysis.** The data were analyzed using GraphPad Prism 7 software. All values are presented as mean ± SD. Multiple comparisons were analyzed by one-way analysis of variance followed by the Bonferroni multiple comparison test. When comparing two groups, unpaired Student's $t$-test or Mann–Whitney test was performed. Differences were considered significant at $P < 0.05$.

**Reporting summary.** Further information on research design is available in the Nature Research Reporting Summary linked to this article.

## Data availability
All data supporting the findings of this study are available within the article and its Supplementary Information files or from the corresponding author upon reasonable request. The source data underlying Fig. 1b, c, d, h, 2a–c, e, g, i, k, 3b–d, f–i, k, 4a, b, d, f, h, i, k, m–p, 5b, d, f, h, j, l, 6a–c, e, f, h, i, k, m–p, 7a, c, e, g–r, and 8a, b, c, d, e, f, g, h, j, l, n and Supplementary Figs. 2b–f, h, 4a, b, d, f, 5a–c, i–l, n, 6b, d–h, 7a, c, e, g, 8a–e, 10a–i, 11a–c and 5d are provided as a Source Data file.

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

## Acknowledgements

This work was supported by grants from the National Key Research and Development Program of China, Ministry of Science and Technology of China (2016YFC1300500-501 and 2016YFC1300500-502), the National Natural Science Foundation of China (General Program 81671156, 31872777, and 81873744, Key Program 81530034), and the Shanghai Municipal Science and Technology Major Project (2018SHZDZX01) and ZJLab.

## Author contributions

L.K., H.Y., and X.Y. performed experiments and analyzed the data. Y.Z., X.B., R.W., Y.C., H.X., H L, L.L., M.-J.S., and Y.T. performed experiments. W.F. and B.Q.Z. supervised the project. L.K., H.Y., W.F., and B.Q.Z. designed the experiments and wrote the manuscript.

## Competing interests

The authors declare no competing interests.
