## [Peer Review File · Nature Communications]

Reviewers' comments:

Reviewer #1 (Remarks to the Author):

This study describes an effect of neutrophils and neutrophil extracellular traps impairing blood-brain barrier breakdown and neovascularization 14 days after brain ischemia induced by permanent occlusion of the middle cerebral artery in mice. The provided information is new since most effects of neutrophils are studied in the acute phase of stroke.

Main comment

The study uses several strategies to promote or inhibit NET formation and it is quite extensive in this regard. Neutrophils are depleted by long-term administration of anti-Ly6G antibodies, the damaging effect of DNA is neutralized by administration of DNase 1, and histone-3 citrullination, a hallmark of NET formation, is promoted or inhibited by gain or loss of function of PAD4, an enzyme that is critical in this process. The main problem is that cells other than neutrophils show H3cit after brain injury. Therefore the specificity of the strategy is incomplete since actions of H3cit and PAD4 on cells other than neutrophils cannot be excluded. Overall, the study demonstrates that extracellular DNA and PAD4 activity promote vascular damage after ischemic stroke and suggest a contribution of NETs in this effect.

Specific comments

1. Mice were treated with anti-Ly6G antibodies for 14 days to deplete neutrophils. What was the status of the mice after this long period of neutrophil depletion? Were other blood leukocytes affected? It is important to show counts of other blood cells to ensure that the observed effects are fully attributable to neutrophil depletion.
2. Did any of these mice with long-term neutrophil depletion die after stroke? Did neutrophil depletion reduce the size of the brain lesion?
3. Did neutrophil depletion in the circulation fully abrogate the presence of neutrophils in the ischemic brain tissue?
4. The precise zone from where IgG extravasation and capillary length were measured should be shown. It is important to clarify precisely the criteria used to distinguish the periphery from the core of infarction in the brain sections and in vivo microscopy? Did measures in the periphery include measures in dorsal and ventral cortical areas flanking the core of the lesion?
5. The presence of extracellular DNA, as assessed with Sytox green in Fig. 3l, m, does not demonstrate that it originated from neutrophils.
6. Fig 3j clearly shows neutrophils with H3cit, in agreement with previous studies. It also shows the presence of H3cit in other cells. For this reason, any treatment intending to inhibit H3cit formation will not necessarily target neutrophils only, but also other cells undergoing histone citrullination.
7. The beneficial effects of DNase 1 treatment are in agreement with a previous report in ischemic mice (de Meyer et al., 2012). This study should be cited. As mentioned above, this treatment is not specific for neutrophil NETs since other cells may release DNA after injury.
8. There is the same problem of cellular specificity with PAD4. While PAD4 is critical for NET formation, the enzyme may also function in other cells, including neurons (e.g. Acharya et al., 2012; Tanikawa et al., 2018).
9. Injection of PAD4 adenovirus in ischemic mice induced NET formation (Fig. 5d). However, this figure shows increased H3cit, as assessed by western blotting, i.e. it does not show that it occurs in neutrophils. The effect of PAD4 overexpression in neutrophils and other cells should be studied at the cellular level.
10. The demonstration of lower numbers of H3cit+ neutrophils after ischemia in PAD4-/- mice as assessed by cell counting is shown in Supplementary Fig. 5b. This result is important and should be shown in the main figures. Presumably PAD4-/- mice also showed less H3cit in cells other than neutrophils. What would the contribution of these other cells be?
11. It would be important to know whether the lesion size was reduced in PAD4-/- mice to exclude long-term differences due to smaller lesions. Please show whether PAD4-deficiency alters the

number of infiltrating neutrophils.

12. The signalling part mediated by STING is again related to PAD4 rather than specifically neutrophils. Did neutrophil depletion downregulate the STING pathway?

References:

Acharya NK, Nagele EP, Han M, Coretti NJ, DeMarshall C, Kosciuk MC, Boulos PA, Nagele RG. Neuronal PAD4 expression and protein citrullination: possible role in production of autoantibodies associated with neurodegenerative disease. *J Autoimmun.* 2012 Jun; 38(4): 369-80.

De Meyer SF, Suidan GL, Fuchs TA, Monestier M, Wagner DD. Extracellular chromatin is an important mediator of ischemic stroke in mice. *Arterioscler Thromb Vasc Biol.* 2012 Aug; 32(8): 1884-91.

Tanikawa C, Ueda K, Suzuki A, Iida A, Nakamura R, Atsuta N, Tohno G, Sobue G, Saichi N, Momozawa Y, Kamatani Y, Kubo M, Yamamoto K, Nakamura Y, Matsuda K. Citrullination of RGG Motifs in FET Proteins by PAD4 Regulates Protein Aggregation and ALS Susceptibility. *Cell Rep.* 2018; 22(6): 1473-1483.

Reviewer #2 (Remarks to the Author):

The authors investigated the effect NETosis on vascular remodeling during stroke recovery. Using a mouse model of distal middle cerebral artery branch occlusion, the authors showed that NETosis primarily occurred in peri-infarct cortex starting 3 days and persisting to 7 days after stroke, which was associated with impairment of cerebral microvascular remodeling assayed by FITC-dextran perfused vessels 14 days after stroke. Blockage of PAD4 by genetic and pharmacological approaches reduced NETosis and increased plasma-perfused microvessels, while administration of DNase 1 also decreased NETosis and augmented microvessels. Furthermore, the authors showed that the STING signaling pathway was involved NETosis-impaired vascular remodeling and BBB leakage by suppressing the type I IFN receptor subunit 1 with neutralized antibody or by STING shRNA. The authors concluded that NETosis mediates cerebral vascular remodeling during stroke recovery. The study is comprehensive including in vitro and in vivo experiments. Factors and signaling pathways investigated in the present study have been previously demonstrated to the development of NETosis¹⁻². Blockage of NETosis improves neurological outcome in experimental stroke. The presence of NETosis has previously been demonstrated in thrombi acquired from patients with acute ischemic stroke who have undergone mechanical thrombectomy³. The novelty of this manuscript is that NETosis mediates cerebral vascular remodeling during stroke recovery; however, this was not supported by the authors' data. Interventions such as DNase 1 and neutralized antibodies used to study the cause effect were initiated 24h after stroke, which likely reduces NETosis and improves cerebral vascular integrity and patency, consequently leading to better vascular remodeling during stroke recovery. Therefore, it is uncertain that NETosis regulates vascular remodeling during repair processes. Below are additional comments.

1) Fig. 1B shows that stroke significantly increased neutrophils in the brain starting 1 day after stroke and there were no significant differences during 1 to 14 days after stroke, suggesting that neutrophils contribute to neurovascular damage, which previously has been well demonstrated.

2) Data of efficiency of PAD4 expression and cellular distribution are required after intra parenchymal injection of recombinant PAD4 adenovirus.

3) The antibody against Ly6G used at 100 µg every second day for 14 days could induce off target effects.

4) Additional data are required to demonstrate that NETosis specifically mediates functional angiogenesis in ischemic brain.

Reference:

1. Fuchs TA, Brill A, Duerschmied D, Schatzberg D, Monestier M, Myers DD, Jr., et al. Extracellular DNA traps promote thrombosis. *Proceedings of the National Academy of Sciences of the United States of America.* 2010; 107: 15880-15885

2. Martinod K, Wagner DD. Thrombosis: Tangled up in nets. *Blood*. 2014;123:2768-2776
3. Laridan E, Denorme F, Desender L, François O, Andersson T, Deckmyn H, et al. Neutrophil extracellular traps in ischemic stroke thrombi. *Annals of Neurology*. 2017;82:223-232

Reviewer #3 (Remarks to the Author):

The authors investigate the role of neutrophils and NETs during ischemic stroke and in the vascular remodeling that occurs afterward. They delineate the time course of neutrophil infiltration as well as NET biomarker presence. Additionally, they see that peripheral blood neutrophils have a higher propensity to release NETs upon additional stimulation. The neutrophil influx in this stroke mouse model is striking and the data regarding DNase administration or PAD4 deficiency are clear in terms of improved vascular remodeling. Neutrophil depletion also had a clear protective effect regarding BBB breakdown and neovascularization. However, the interpretation of NETs being the reason behind these effects needs some additional investigation, particularly regarding PAD4 which is also expressed by cells within the brain. My comments are as follows:

Major:

- 1- PAD4 is also expressed by cells in the brain, and therefore protective results in the PAD4-knockouts or upon PAD inhibition are not necessarily due to NETs. A neutrophil-targeted approach, or an experiment with DNase administration in PAD4^{-/-} mice, would be informative in this respect.
- 2- PAD4 overexpression by injection into the brain may also upregulate PAD4 expression within the brain. This experiment performed with infusion of isolated neutrophils with PAD4-overexpressed would provide a more clear result.
- 3- How was Cl-amidine solution prepared for in vivo use? Low solubility of Cl-amidine in an aqueous solution such as saline generally requires prior dissolving in DMSO or EtOH. If this was the case, then the proper vehicle control should be used for these experiments.
- 4- The use of Student's t-test for comparison of 2 groups should be justified for all data sets with normality tests, otherwise a non-parametric test is needed. As the results are all shown as bar graphs rather than with individual data points, it is unclear if use of Student's t-test is acceptable throughout the entire manuscript.
- 5- The description for validation of neutrophil depletion in the methods is listed as using anti-7/4 antibody, yet Supplemental figure 2 shows Ly6G FACS plots. Which antibody was used to verify neutrophil depletion? And which gating was used to quantify the Ly6G⁺ cells in Supp Figure 2b? This is not clear from the figure legend. A clear validation of neutrophil depletion using 7/4 rather than the same antibody used for depletion is needed.
- 6- What is the difference between panels n and o in Figure 5? And also for panels l, m in Figure 6? This is not clear from the figure legend text.
- 7- Unless quantification has been performed for figures 6e and 6h, then a conclusion regarding increase should not be made in the text. What number of samples are these blots representative of?

Minor:

- 1- Description of how electrocoagulation was performed should be added to the methods in order to allow researchers to follow up on this study.

2- Pg. 19, first line of the concluding paragraph: "stroke caused neutrophils accumulated in the brain" would be better worded as "stroke caused neutrophil accumulation in the brain"

3- Cl-amidine is not a specific PAD4 inhibitor but rather an inhibitor of several PAD family members. References in the text to PAD4 inhibition by Cl-amidine should rather say "PAD inhibition"

Point by point responses to the Reviewers' comments.

Reviewer #1 (Remarks to the Author):

This study describes an effect of neutrophils and neutrophil extracellular traps impairing blood-brain barrier breakdown and neovascularization 14 days after brain ischemia induced by permanent occlusion of the middle cerebral artery in mice. The provided information is new since most effects of neutrophils are studied in the acute phase of stroke.

Main comment

The study uses several strategies to promote or inhibit NET formation and it is quite extensive in this regard. Neutrophils are depleted by long-term administration of anti-Ly6G antibodies, the damaging effect of DNA is neutralized by administration of DNase 1, and histone-3 citrullination, a hallmark of NET formation, is promoted or inhibited by gain or loss of function of PAD4, an enzyme that is critical in this process. The main problem is that cells other than neutrophils show H3cit after brain injury. Therefore the specificity of the strategy is incomplete since actions of H3cit and PAD4 on cells other than neutrophils cannot be excluded. Overall, the study demonstrates that extracellular DNA and PAD4 activity promote vascular damage after ischemic stroke and suggest a contribution of NETs in this effect.

Specific comments

1. Mice were treated with anti-Ly6G antibodies for 14 days to deplete neutrophils. What was the status of the mice after this long period of neutrophil depletion? Were other blood leukocytes affected? It is important to show counts of other blood cells to ensure that the observed effects are fully attributable to neutrophil depletion.

At 14 days after stroke, no significant differences were observed in body weight (24.01 ± 0.91 g versus 24.05 ± 0.70 g; $P = 0.90$, $n = 12$) and survival rate (91.7% versus 91.7%) between neutrophil depletion and control groups. These results are now included in the Methods section page 26.

We agree that these analyses are very important to document. As suggested, we have performed peripheral blood counts. This analysis showed that anti-Ly6G treatment reduced neutrophil counts in the blood by approximately 80% when compared with the isotype control group (Fig. 2a). White blood cell counts were also reduced (Fig. 2b), whereas red blood cell, platelet, monocyte and lymphocyte counts were not significantly affected by anti-Ly6G treatment (Supplementary Fig. 2c-f), as reported (Stirling DP, et al. J Neurosci. 2009; Thayer TC, et al. Diabetes. 2011). These results are now included in the Results section page 5, line 16-19, the Methods section page 27, line 19-21, and are presented in a Supplementary Figure 2c-f.

2. Did any of these mice with long-term neutrophil depletion die after stroke? Did neutrophil depletion reduce the size of the brain lesion?

One out of twelve mice died in neutrophil depletion and control groups during 14 days after stroke.

We agree with the reviewer and have now quantified infarct volumes in anti-Ly6G-treated and control IgG-treated mice. Our data showed that there was no significant difference in infarct volume between anti-Ly6G-treated and control IgG-treated mice at 14 days

(Supplementary Fig. 2g, h). These results are now included in the Results section page 6, line 7-8, the Methods section page 24, line 15-17.

3. Did neutrophil depletion in the circulation fully abrogate the presence of neutrophils in the ischemic brain tissue?

As suggested by the reviewer, we have performed new studies to determine the effects of neutrophil depletion on the presence of neutrophils in the ischemic brain tissue. Immunostaining analysis indicated that the number of infiltrating neutrophils in the ischemic brain was significantly less in anti-Ly6G -treated mice than in control IgG-treated mice (Fig. 2c). We have added these results in the Results section page 5, line 19-21.

4. The precise zone from where IgG extravasation and capillary length were measured should be shown. It is important to clarify precisely the criteria used to distinguish the periphery from the core of infarction in the brain sections and in vivo microscopy? Did measures in the periphery include measures in dorsal and ventral cortical areas flanking the core of the lesion?

We agree with the reviewer and have now provided a schematic representation of a hematoxylin and eosin-stained coronal section shows the zone used for measurements of IgG extravasation, capillary length and in vivo multiphoton microscopy in the peri-infarct cortical areas (Fig. 1a).

5. The presence of extracellular DNA, as assessed with Sytox green in Fig. 3l, m, does not demonstrate that it originated from neutrophils.

We have now provided representative multiphoton microscopy images of neutrophils and extracellular DNA in Fig. 3m. Extracellular DNA (green) were labeled with intravenous injection of Sytox green, and neutrophils (red) with intravenous injection of PE-conjugated monoclonal Ly6G antibody.

6. Fig 3j clearly shows neutrophils with H3Cit, in agreement with previous studies. It also shows the presence of H3Cit in other cells. For this reason, any treatment intending to inhibit H3Cit formation will not necessarily target neutrophils only, but also other cells undergoing histone citrullination.

The referee raises a most critical issue. To identify which type of cells expressed H3Cit after stroke, double immunofluorescence with confocal microscopy was performed on brain sections. This analysis revealed that H3Cit was colocalized with Ly6G-positive neutrophils, F4/80-positive macrophages/microglial cells, Iba1-positive microglial cells, NeuN-positive neurons, and GFAP-positive astrocytes (Fig. 3k). Importantly, 78.7% of the H3Cit-positive cells were Ly6G-positive neutrophils. This prompted us to investigate the importance of neutrophil NETs in vascular remodeling. Our results showed that treatment with DNase 1 in combination with anti-Ly6G antibody did not further improve neovascularization and BBB leakage compared with mice treated with anti-Ly6G antibody alone (Fig. 4n-p). These data indicate that DNase I primarily digests NETs generated by neutrophils and that neutrophil NETs play a crucial role in vascular remodeling after stroke. However, in addition to neutrophil NETs, H3Cit in other cells may also contribute to the impaired vascular remodeling. These results are now included in the Results section page 8, line 7-12 and page 11, line 1-7.

7. The beneficial effects of DNase 1 treatment are in agreement with a previous report in ischemic mice (de Meyer et al., 2012). This study should be cited. As mentioned above, this treatment is not specific for neutrophil NETs since other cells may release DNA after injury.

We have now cited this work (De Meyer SF, et al. Arterioscler Thromb Vasc Biol) in the Discussion section on page 22, line 20-21. Thank you for pointing out this publication.

As discussed above and shown in Fig. 3k and Fig. 4 n-p, we found that H3Cit was mainly colocalized with neutrophils after stroke (Fig. 3k). Our results also showed that treatment with DNase 1 in combination with anti-Ly6G antibody did not further improve neovascularization and BBB leakage compared with mice treated with anti-Ly6G antibody alone (Fig. 4n-p). These data indicate that DNase I primarily digests NETs generated by neutrophils after stroke. However, in addition to neutrophil NETs, H3Cit in other cells may also contribute to the impaired vascular remodeling. These results are now included in the Results section page 8, line 7-12 and page 11, line 1-7.

8. There is the same problem of cellular specificity with PAD4. While PAD4 is critical for NET formation, the enzyme may also function in other cells, including neurons (e.g. Acharya et al., 2012; Tanikawa et al., 2018).

The referee asks an important question. This question was also raised by Referee 3. We have now added new experiments to try to address this important issue. Previous studies have shown that PAD4 may also function in other cells (Acharya NK, et al. J Autoimmun. 2012; Tanikawa C, et al. Cell Rep. 2018). We found that PAD4 deficiency substantially reduced neutrophil NETs, as seen by the decrease in H3Cit⁺ neutrophils (Fig. 7d, e). However, PAD4 deficiency did not significantly affect the number of H3Cit⁺ macrophages/microglial cells, H3Cit⁺ microglial cells, H3Cit⁺ neurons, and H3Cit⁺ astrocytes (Supplementary Fig. 6d-h). Furthermore, we found that PAD4 deficiency had no beneficial effect on BBB permeability ($1.18 \times 10^4 \pm 2702$ versus $1.10 \times 10^4 \pm 2697$, $P = 0.66$, $n = 4-6$), microvascular length (12.8 ± 1.68 versus 13.82 ± 2.31 , $P = 0.42$, $n = 4-6$), and perfused capillary length (1871 ± 136 versus 1978 ± 91 , $P = 0.21$, $n = 4-6$) in mice subjected to anti-Ly6G antibody treatment. These findings suggest that PAD4 may primarily mediate the formation of neutrophil NETs after stroke. These results are now included in the text page 16, line 7-9 and page 16, line 25.

9. Injection of PAD4 adenovirus in ischemic mice induced NET formation (Fig. 5d). However, this figure shows increased H3Cit, as assessed by western blotting, i.e. it does not show that it occurs in neutrophils. The effect of PAD4 overexpression in neutrophils and other cells should be studied at the cellular level.

We agree with the reviewer and have done the experiment as suggested. Immunohistochemical quantification revealed a significant increase in the number of H3Cit⁺ neutrophils (Fig. 6f), whereas the number of H3Cit⁺ macrophages/microglial cells, H3Cit⁺ microglial cells, H3Cit⁺ neurons, and H3Cit⁺ astrocytes were not affected by PAD4 adenovirus treatment (Supplementary Fig. 5i-l). These results are now included in the text page 13, line 25-26 and page 14, line 1-2.

10. The demonstration of lower numbers of H3Cit+ neutrophils after ischemia in PAD4-/- mice as assessed by cell counting is shown in Supplementary Fig. 5b. This result is important and should be shown in the main figures. Presumably PAD4-/- mice also showed less H3Cit in cells other than neutrophils. What would the contribution of these other cells be?

We agree with the reviewer and have put Supplementary Fig. 5b in new Fig. 7e.

As suggested by the reviewer, we have performed new experiments to determine whether PAD4^{-/-} mice also showed less H3Cit in cells other than neutrophils. We found that PAD4 deficiency substantially reduced neutrophil NETs, as seen by the decrease in H3Cit⁺ neutrophils (Fig. 7d, e). However, PAD4 deficiency did not significantly affect the number of

H3Cit⁺ macrophages/microglial cells, H3Cit⁺ microglial cells, H3Cit⁺ neurons, and H3Cit⁺ astrocytes (Supplementary Fig. 6d-h). These data are presented in the Results section page 16, line 7-9.

11. It would be important to know whether the lesion size was reduced in PAD4^{-/-} mice to exclude long-term differences due to smaller lesions. Please show whether PAD4-deficiency alters the number of infiltrating neutrophils.

We agree with the reviewer and have now quantified infarct volumes in WT and PAD4^{-/-} mice. Our data showed that there was no significant difference in ischemic lesion between WT and PAD4^{-/-} mice at 14 days (Supplementary Fig. 6a, b). Thus, this approach allowed us to provide evidence that effects of PAD4 deficiency on long-term outcomes are not secondary to the reduced lesion size. These results are now included in the Results section page 15, line 21-23, the Methods section page 24, line 15-17.

As suggested by the reviewer, we detected whether PAD4 deficiency alters the number of infiltrating neutrophils. We observed that PAD4 deficiency did not alter the number of infiltrating neutrophils in the ischemic brain tissue (Supplementary Fig. 6c, d). These results are now included in the Results section page 15, line 24-26.

12. The signaling part mediated by STING is again related to PAD4 rather than specifically neutrophils. Did neutrophil depletion downregulate the STING pathway?

We agree that these analyses are very important to document. To test whether the signaling part mediated by STING is related to neutrophils, we isolated neutrophils from bone marrow of ischemic mice and stimulated them with LPS in the presence or absence of the PAD inhibitor Cl-amidine. We found that the STING-mediated signals were activated by LPS stimulation and this effect was reversed by the addition of Cl-amidine (Fig. 8g, h; Supplementary Fig. 11a-c). Consistent with these findings, we observed that neutrophil depletion resulted in a significant reduction in the levels of STING, and the STING downstream signaling molecules including pTBK1, pIRF3 and IFN- β in the ischemic brain relative to IgG-treated controls (Fig. 8e, f; Supplementary Fig. 10g-i). Data are presented in Results section page 19, line 13-15 and page 19, line 8-11.

Reviewer #2 (Remarks to the Author):

The authors investigated the effect NETosis on vascular remodeling during stroke recovery. Using a mouse model of distal middle cerebral artery branch occlusion, the authors showed that NETosis primarily occurred in peri-infarct cortex starting 3 days and persisting to 7 days after stroke, which was associated with impairment of cerebral microvascular remodeling assayed by FITCdextran perfused vessels 14 days after stroke. Blockage of PAD4 by genetic and pharmacological approaches reduced NETosis and increased plasma-perfused microvessels, while administration of DNase 1 also decreased NETosis and augmented microvessels. Furthermore, the authors showed that the STING signaling pathway was involved NETosis-impaired vascular remodeling and BBB leakage by suppressing the type I IFN receptor subunit 1 with neutralized antibody or by STING shRNA. The authors concluded that NETosis mediates cerebral vascular remodeling during stroke recovery. The study is comprehensive including in vitro and in vivo experiments. Factors and signaling pathways investigated in the present study have been previously demonstrated to the development of NETosis¹⁻². Blockage of NETosis improves neurological outcome in experimental stroke. The presence of NETosis has previously been demonstrated in thrombi acquired from patients with acute ischemic stroke who have undergone mechanical thrombectomy³. The novelty of this manuscript is that NETosis mediates cerebral vascular remodeling during stroke recovery; however,

this was not supported by the authors' data. Interventions such as DNase 1 and neutralized antibodies used to study the cause effect were initiated 24h after stroke, which likely reduces NETosis and improves cerebral vascular integrity and patency, consequently leading to better vascular remodeling during stroke recovery. Therefore, it is uncertain that NETosis regulates vascular remodeling during repair processes.

As correctly suggested by the reviewer, we have referenced the articles documenting the role of NETosis in thrombosis and the presence of NETosis in patients with acute ischemic stroke (Fuchs, TA. et al. Proc Natl Acad Sci U S A 2010; Martinod K, Wagner DD. Blood. 2014; Laridan E, et al. Ann Neurol 2017) in the Introduction page 2, line 17 and page 2, line 18.

This is an important comment by the reviewer that we felt we needed to explore carefully. We have performed new studies to detect whether NETosis regulates vascular remodeling during repair processes. We found that injection of anti-Ly6G antibody beginning 7 days after stroke reduced extravascular IgG deposits at 14 days (new Fig. 5a, b). Furthermore, we observed significant increases in microvascular length (Fig. 5e, f) and perfused cortical vessels (Fig. 5i, j) in anti-Ly6G antibody-treated mice compared with control IgG-treated mice. Treatment with DNase 1 at 7 days after stroke also attenuated BBB disruption (Fig. 5c, d), increased microvessels (Fig. 5g, h) and improved capillary perfusion (Fig. 5k, l) at 14 days. These results are now included in the Results section page 12, line 9-16.

Below are additional comments

1) Fig. 1B shows that stroke significantly increased neutrophils in the brain starting 1 day after stroke and there were no significant differences during 1 to 14 days after stroke, suggesting that neutrophils contribute to neurovascular damage, which previously has been well demonstrated.

As suggested, we have now cited these previous findings in the Results section on page 3, line 13.

2) Data of efficiency of PAD4 expression and cellular distribution are required after intra parenchymal injection of recombinant PAD4 adenovirus.

As suggested by the reviewer, we have now examined the efficiency of PAD4 expression and cellular distribution after intra parenchymal injection of recombinant PAD4 adenovirus. Immunohistochemical analysis indicated extensive expression of recombinant Adeno-PAD4-flag-infected cells in the cortex at 4 days after injection (Fig. 6d). Overexpression of PAD4 in the cortex was also confirmed by immunoblotting (Supplementary Fig. 5a, b). PAD4-flag was present in neurons, neutrophils, macrophages/microglial cells and microglial cells, but was rarely detected in astrocytes (Supplementary Fig. 5d-h). We then observed a significant increase in the number of H3Cit⁺ neutrophils (Fig. 6f), whereas the number of H3Cit⁺ macrophages/microglial cells, H3Cit⁺ microglial cells, H3Cit⁺ neurons, and H3Cit⁺ astrocytes were not affected by PAD4 adenovirus treatment (Supplementary Fig. 5i-l). These data are presented in a new supplemental Fig. 5, and the Results section page 13, line 18-26.

3) The antibody against Ly6G used at 100 µg every second day for 14 days could induce off target effects.

The dose of anti-Ly6G antibody and the length of treatment were chosen based on previous studies (Drechsler M, et al. Circulation. 2010). At 14 days after stroke, no significant differences were observed in body weight (24.01 ± 0.91 g versus 24.05 ± 0.70 g; P = 0.90, n = 12) and survival rate (91.7% versus 91.7%) between neutrophil depletion and control groups. Interestingly, in Alzheimer's disease models, depletion of neutrophils by 300 µg of anti-Ly6G antibody every second day for 1 month reduced Alzheimer's disease pathogenesis and

improved memory (Zenaro E, et al. Nat Med. 2015). However, anti-integrin therapies were previously shown to induce progressive multifocal leukoencephalopathy in patients with autoimmune disorders (Baldwin KJ & Hogg JP. Curr Opin Neurol. 2013). Further investigation of whether this long period of neutrophil depletion can induce side effects will be essential. We have included these explanations in the Method section, page 26, line 8-10 and the Discussion section, page 22, line 6-11.

4) Additional data are required to demonstrate that NETosis specifically mediates functional angiogenesis in ischemic brain.

We agree that these analyses are very important to document. As suggested, we evaluated functional angiogenesis by in vivo administration of tomato-lectin. This experiment showed that the total length of tomato-lectin-perfused vessels were increased in the ischemic cortex in DNase 1-treated mice (Fig. 4l, m), PAD inhibitor Cl-amidine-treated mice (Fig. 7l; Supplementary Fig. 7h) and PAD4^{-/-} mice (Fig. 7l; Supplementary Fig. 7h) compared with the control group. These results are now included in the Results section page 10, line 24-25 and page 16, line 19-20.

Reviewer #3 (Remarks to the Author):

The authors investigate the role of neutrophils and NETs during ischemic stroke and in the vascular remodeling that occurs afterward. They delineate the time course of neutrophil infiltration as well as NET biomarker presence. Additionally, they see that peripheral blood neutrophils have a higher propensity to release NETs upon additional stimulation. The neutrophil influx in this stroke mouse model is striking and the data regarding DNase administration or PAD4 deficiency are clear in terms of improved vascular remodeling. Neutrophil depletion also had a clear protective effect regarding BBB breakdown and neovascularization. However, the interpretation of NETs being the reason behind these effects needs some additional investigation, particularly regarding PAD4 which is also expressed by cells within the brain. My comments are as follows:

Major:

1- PAD4 is also expressed by cells in the brain, and therefore protective results in the PAD4-knockouts or upon PAD inhibition are not necessarily due to NETs. A neutrophil-targeted approach, or an experiment with DNase administration in PAD4^{-/-} mice, would be informative in this respect.

We agree that these analyses are very important to document. This question was also raised by Referee 1. As suggested, we have performed these new experiments and the results of which are shown in Fig. 7m-o. We found that treatment with anti-Ly6G antibody or DNase 1 had no beneficial effect in PAD4^{-/-} mice on BBB permeability (Fig. 7m), microvascular length (Fig. 7n), and perfused capillary length (Fig. 7o). These data indicate that the effects of PAD4 deficiency on neovascularization and vascular remodeling are due to NETs. These results are now included in the text page 16, line 20-23.

2- PAD4 overexpression by injection into the brain may also upregulate PAD4 expression within the brain. This experiment performed with infusion of isolated neutrophils with PAD4-overexpressed would provide a more clear result.

The suggestion of infusing isolated neutrophils with PAD4-overexpressed is interesting. Since the short life span of neutrophils limits the efficiency of transfection, infusion of isolated

neutrophils with PAD4-overexpressed is difficult. As suggested, the cellular localization of overexpressed PAD4 was determined using double immunostaining. We found that Padi4-flag was present in neurons, neutrophils, macrophages/microglial cells and microglial cells, but was rarely detected in astrocytes (Supplementary Fig. 5d-h) after injection of recombinant PAD4 adenovirus. Importantly, we observed a significant increase in the number of H3Cit⁺ neutrophils (Fig. 6f), whereas the number of H3Cit⁺ macrophages/microglial cells, H3Cit⁺ microglial cells, H3Cit⁺ neurons, and H3Cit⁺ astrocytes were not affected by PAD4 adenovirus treatment (Supplementary Fig. 5i-l). These data suggest that overexpression of PAD4 primarily increased the formation of neutrophil NETs after stroke. We have added these data in the Results section on page 13, line 21-23 and page 13, line 25.

3- How was Cl-amidine solution prepared for in vivo use? Low solubility of Cl-amidine in an aqueous solution such as saline generally requires prior dissolving in DMSO or EtOH. If this was the case, then the proper vehicle control should be used for these experiments.

We are sorry for the mistake. Stock solution of PAD inhibitor Cl-amidine (506282, Millipore) was dissolved in DMSO (Sigma-Aldrich). The stock solution was dissolved in saline (5% v/v) and injected i.p. at 10 mg/kg 24 hours after cerebral ischemia and then every day until mice were sacrificed. Vehicle (saline containing 5% DMSO) was administered in the same way. These sentences are added in the Method section, page 26, line 22-26.

4- The use of Student's t-test for comparison of 2 groups should be justified for all data sets with normality tests, otherwise a nonparametric test is needed. As the results are all shown as bar graphs rather than with individual data points, it is unclear if use of Student's t-test is acceptable throughout the entire manuscript.

We are sorry for the mistake in analysis and thank the reviewer for catching it. We have reexamined the data. When the data were not normally distributed, the Student's t-test have now been replaced by the Mann-Whitney test (Fig. 3c, new Fig. 8c). We have replaced "the Student's t-test" by "the Mann-Whitney test" in the figure legends and Statistical analysis part.

5- The description for validation of neutrophil depletion in the methods is listed as using anti-7/4 antibody, yet Supplemental figure 2 shows Ly6G FACS plots. Which antibody was used to verify neutrophil depletion? And which gating was used to quantify the Ly6G+ cells in Supp Figure 2b? This is not clear from the figure legend. A clear validation of neutrophil depletion using 7/4 rather than the same antibody used for depletion is needed.

We apologize for the confusion. The monoclonal anti-mouse Ly6G antibody (1A8 clone; BE0075-1, BioXCell) was used for neutrophil depletion. To verify neutrophil depletion, blood neutrophil levels were evaluated by flow cytometry using a FITC-conjugated rat monoclonal anti-mouse neutrophil antibody (anti-7/4; ab53453, Abcam). We have now added these in the Method section, page 26, line 3-4 and line 10-13.

6- What is the difference between panels n and o in Figure 5? And also for panels l, m in Figure 6? This is not clear from the figure legend text.

We apologize for the confusion. Panel n in Figure 5 is the analysis of forelimb foot faults and panel o is hindlimb foot faults. We have now added these in the new Figure 6n, o and Figure 7p, q.

7- Unless quantification has been performed for figures 6e and 6h, then a conclusion regarding increase should not be made in the text. What number of samples are these

blots representative of?

We agree with the reviewer and have now provided the quantitative data for figures 6e and 6h (Supplementary Fig. 8a-e)

Minor:

1- Description of how electrocoagulation was performed should be added to the methods in order to allow researchers to follow up on this study.

We have added more detail information in the Method parts on page 24.

2- Pg. 19, first line of the concluding paragraph: “stroke caused neutrophils accumulated in the brain” would be better worded as “stroke caused neutrophil accumulation in the brain”

We agree with the reviewer and have now replaced “stroke caused neutrophils accumulated in the brain” by “stroke caused neutrophil accumulation in the brain”.

3- Cl-amidine is not a specific PAD4 inhibitor but rather an inhibitor of several PAD family members. References in the text to PAD4 inhibition by Cl-amidine should rather say “PAD inhibition”

We agree with the reviewer and have now replaced “PAD4 inhibition” by “PAD inhibition” throughout the text.

We thank the reviewers for helpful suggestions and careful reading of the manuscript. We hope that the added explanations and the new results have improved our manuscript.

Sincerely,

Bing-Qiao Zhao & Wenying Fan, MD, PhD

State Key Laboratory of Medical Neurobiology
Fudan University
138 Yixueyuan Road, Shanghai 200032, China

References:

1. Stirling DP, Liu S, Kubes P, Yong VW. Depletion of Ly6G/Gr-1 leukocytes after spinal cord injury in mice alters wound healing and worsens neurological outcome. *J Neurosci*. 2009;29:753-764.
2. Thayer TC, Delano M, Liu C, Chen J, Padgett LE, Tse HM, Annamali M, Piganelli JD, Moldawer LL, Mathews CE. Superoxide production by macrophages and T cells is critical for the induction of autoreactivity and type 1 diabetes. *Diabetes*. 2011;60:2144-2151.
3. De Meyer SF, Suidan GL, Fuchs TA, Monestier M, Wagner DD. Extracellular chromatin is an important mediator of ischemic stroke in mice. *Arterioscler Thromb Vasc Biol*. 2012;32:1884-1891.
4. Acharya NK, et al. Neuronal PAD4 expression and protein citrullination: possible role in production of autoantibodies associated with neurodegenerative disease. *J Autoimmun*. 2012;38:369-380.
5. Tanikawa C, et al. Citrullination of RGG Motifs in FET Proteins by PAD4 Regulates Protein Aggregation and ALS Susceptibility. *Cell Rep*. 2018;22:1473-1483.

6. Fuchs, T. A. et al. Extracellular DNA traps promote thrombosis. *Proc Natl Acad Sci U S A*. 2010;107,15880-15885.
7. Martinod K, Wagner DD. Thrombosis: Tangled up in nets. *Blood*. 2014;123:2768-2776.
8. Laridan, E. et al. Neutrophil extracellular traps in ischemic stroke thrombi. *Ann Neurol* 2017;82,223-232.
9. Drechsler M, Megens RT, van Zandvoort M, Weber C, Soehnlein O. Hyperlipidemia-triggered neutrophilia promotes early atherosclerosis. *Circulation*. 2010;122:1837-1845.
10. Zenaro E, et al. Neutrophils promote Alzheimer's disease-like pathology and cognitive decline via LFA-1 integrin. *Nat Med*. 2015;21:880-886.
11. Baldwin KJ, Hogg JP. Progressive multifocal leukoencephalopathy in patients with multiple sclerosis. *Curr Opin Neurol*. 2013;26:318-323.

Reviewers' comments:

Reviewer #1 (Remarks to the Author):

I believe the manuscript is improved. I have some minor remarks:

The time point of each experiment should be clearly stated in the figure legends.

In Methods, the time '7 days' was added to the section 'DNase I treatment' (page 26). However, the sentence is now confusing. Please check and explain properly since now the treatment regimens are not clear.

Referring to treatment with DNAase (main text page 12) it is stated that 'Treatment with DNase I at 7 days after stroke...'. However, according to Methods the treatment seems to include injections every 12h, so the text probably means 'starting at 7 days'. Please clarify.

Fig. 6d is meaningless without the appropriate controls.

In page 19, regarding the ex vivo LPS treatment of neutrophils, it is stated that 'STING-mediated signals were activated by LPS stimulation and this effect was reversed by the addition of CI-amidine'. Considering that CI-amidine was added at the same time than LPS, CI-amidine did not reverse, but prevented or rather attenuated the effect of LPS.

Reviewer #2 (Remarks to the Author):

The authors have performed additional experiments to address my concerns and the manuscript has been improved. Additional new data support the author's conclusion. No further comments.

Reviewer #3 (Remarks to the Author):

All of my comments have been sufficiently addressed with this revised manuscript.

Reviewers' comments:

Reviewer #1 (Remarks to the Author):

I believe the manuscript is improved. I have some minor remarks:

The time point of each experiment should be clearly stated in the figure legends.

In Methods, the time '7 days' was added to the section 'DNase I treatment' (page 26). However, the sentence is now confusing. Please check and explain properly since now the treatment regimens are not clear.

Referring to treatment with DNAase (main text page 12) it is stated that 'Treatment with DNase I at 7 days after stroke...'. However, according to Methods the treatment seems to include injections every 12h, so the text probably means 'starting at 7 days'. Please clarify.

Fig. 6d is meaningless without the appropriate controls.

In page 19, regarding the ex vivo LPS treatment of neutrophils, it is stated that 'STING-mediated signals were activated by LPS stimulation and this effect was reversed by the addition of Cl-amidine'. Considering that Cl-amidine was added at the same time than LPS, Cl-amidine did not reverse, but prevented or rather attenuated the effect of LPS.

Point by point responses to the Reviewers' comments.

Reviewer #1 (Remarks to the Author):

I believe the manuscript is improved. I have some minor remarks:

The time point of each experiment should be clearly stated in the figure legends.

We agree with the reviewer and have added the time point of each experiment in the figure legends.

In Methods, the time '7 days' was added to the section 'DNase I treatment' (page 26). However, the sentence is now confusing. Please check and explain properly since now the treatment regimens are not clear.

We apologize for the confusion. As suggested, we have included an explanation on page 27, line 17.

Referring to treatment with DNAase (main text page 12) it is stated that 'Treatment with DNase I at 7 days after stroke...'. However, according to Methods the treatment seems to include injections every 12h, so the text probably means 'starting at 7 days'. Please clarify.

We agree with the reviewer and have now replaced "Treatment with DNase I at 7 days after stroke..." by "Treatment with DNase I starting at 7 days after stroke...". Thank you for pointing out this error.

Fig. 6d is meaningless without the appropriate controls.

We agree with the reviewer and have now provided a representative image of the control empty adenovirus in Fig. 6d.

In page 19, regarding the ex vivo LPS treatment of neutrophils, it is stated that ‘STING-mediated signals were activated by LPS stimulation and this effect was reversed by the addition of Cl-amidine’. Considering that Cl-amidine was added at the same time than LPS, Cl-amidine did not reverse, but prevented or rather attenuated the effect of LPS.

We agree with the reviewer and have now replaced “STING-mediated signals were activated by LPS stimulation and this effect was reversed by the addition of Cl-amidine” by “STING-mediated signals were activated by LPS stimulation and this effect was prevented by the addition of Cl-amidine”.

We thank the reviewer for carefully reading the manuscript and valuable suggestions. We believe that the review process improved our manuscript.

REVIEWERS' COMMENTS:

Reviewer #1 (Remarks to the Author):

I believe the authors corrected and improved the text.

Reviewers' comments:

Reviewer #1 (Remarks to the Author):

I believe the authors corrected and improved the text.

Point by point responses to the Reviewers' comments:

Reviewer #1 (Remarks to the Author):

I believe the authors corrected and improved the text.

We appreciate that the reviewer is now satisfied with our revised manuscript.

We thank the reviewer for carefully reading the manuscript. We believe that the review process improved our manuscript.

Sincerely,

Bing-Qiao Zhao & Wenying Fan, MD, PhD

State Key Laboratory of Medical Neurobiology
Fudan University
138 Yixueyuan Road, Shanghai 200032, China